# Synergistic Patch Pruning for Vision Transformer: Unifying Intra- & Inter-Layer Patch Importance

**Yuyao Zhang**
Institute of Advanced Technology
University of Science and Technology of China
Hefei, Anhui, 230026, P.R.China
`juttazhang@mail.ustc.edu.cn`

**Lan Wei**
School of Computer Science and Technology
University of Science and Technology of China
Hefei, Anhui, 230026, P.R.China
`weilan@mail.ustc.edu.cn`

**Nikolaos M. Freris** [*]
School of Computer Science and Technology
University of Science and Technology of China
Hefei, Anhui, 230026, P.R.China
`nfr@ustc.edu.cn`

## Abstract

The Vision Transformer (ViT) has emerged as a powerful architecture for various computer vision tasks. Nonetheless, this comes with substantially heavier computational costs than Convolutional Neural Networks (CNNs). The attention mechanism in ViTs, which integrates information from different image patches to the class token ([CLS]), renders traditional structured pruning methods used in CNNs unsuitable. To overcome this issue, we propose SynergisTic pAtch pRuning (STAR) that unifies intra-layer and inter-layer patch importance scoring. Specifically, our approach combines a) online evaluation of intra-layer importance for the [CLS] and b) offline evaluation of the inter-layer importance of each patch. The two importance scores are fused by minimizing a weighted average of Kullback-Leibler (KL) Divergences and patches are successively pruned at each layer by maintaining only the top-$k$ most important ones. Unlike prior art that relies on manual selection of the pruning rates at each layer, we propose an automated method for selecting them based on offline-derived metrics. We also propose a variant that uses these rates as weighted percentile parameters (for the layer-wise normalized scores), thus leading to an alternate adaptive rate selection technique that is input-based. Extensive experiments demonstrate the significant acceleration of the inference with minimal performance degradation. For instance, on the ImageNet dataset, the pruned DeiT-Small reaches a throughput of 4,256 images/s, which is over 66% higher than the much smaller (unpruned) DeiT-Tiny model, while having a substantially higher accuracy (+6.8% Top-1 and +3.1% Top-5).

## 1 Introduction

Transformers have gained popularity as backbone networks for various applications. The key to the proliferation of the Vision Transformer (ViT) and its variants lies in a unified architecture that effectively tackles the long-range dependencies in images (Dosovitskiy et al., 2021). However, the intricate architecture involved brings along intensive computational costs that prohibit the model deployment on resource-constrained devices (Zheng et al., 2022). This motivates the need for ViT compression. Existing research on ViT compression has drawn inspiration from techniques used in Convolutional Neural Networks (CNNs), such as weight pruning (Chen et al., 2021b; Yu & Wu, 2023; Yu et al., 2022; Zhu et al., 2021), quantization (Liu et al., 2021), and compact architecture design (Chen et al., 2021a). There are also several dynamic methods to adjust the pruning thresholds based on the input (Zhang & Freris, 2023; Tang et al., 2021; Lin et al., 2020; Wang et al., 2020). For instance, methods like channel or filter pruning aim to reduce the model size by removing unnecessary structures. However, hasty pruning of entire channels in ViTs can result in information

---
[*]Corresponding author

loss (Hou & Kung, 2022), as ViTs split input images into multiple patches and perform calculations on all patches in parallel. Moreover, the self-attention mechanism in ViT aggregates features using position embedding and assesses patch significance to the class token ([CLS]) in different layers (blocks) of the network. The quadratic time and memory complexity associated with the number of input patches is the source of the main computational burden in ViTs.

For this reason, patch pruning becomes topical. Recent studies (Liang et al., 2022; Tang et al., 2022) indicate that certain patches in ViTs encode task-irrelevant information (such as the background) while the importance of any given patch varies across ViT blocks (see Fig.1). Nonetheless, most existing techniques do not fully explore the redundancies *across* different ViT blocks (Pan et al., 2021a; Xu et al., 2022). It is thus crucial to ensure that patches critical for deeper layers are not mistakenly pruned in shallower layers. Mainstream patch pruning methods (Dong et al., 2023; Wei et al., 2023) employ recursive sampling techniques to select informative patches and reduce the model inference time. However, these 'static' methods (Chen et al., 2021b; Fayyaz et al., 2022; Rao et al., 2021) prune a predetermined subset of patches regardless of input image dependency, limiting their performance. In contrast, input-adaptive approaches Xu et al. (2022), like the one proposed in this paper, perform pruning during inference (adaptive to the input data).

To tackle this, we propose a new method called SynergisTic pAtch pRuning (STAR) that combines dynamic evaluation of intra-layer patch importance during inference with inter-layer patch importance calculated offline (using tools from model interpretability). The former is achieved by using the attention map at each block, while we employ Layer-wise Relevance Propagation (LRP) (Chefer et al., 2021) for the latter. Subsequently, the two importance scores at each layer are fused so as to prune by retaining the most important ones. Given that the scores are positive, the fusion is accomplished by viewing them as a probability mass function (pmf) and seeking to find the pmf that minimizes a weighted sum of the Kullback-Leibler (KL) divergence which admits a simple closed-form solution as the (element-wise) product of the scores raised to two powers that add to one. Apart from assessing patch importance, another crucial challenge is determining the retention rate per layer to balance compression and accuracy. In this regard, existing patch pruning methods fall into two categories. There are methods like Liang et al. (2022); Yang et al. (2021), that rely on empirical per-layer retention rate settings: this process heavily relies on the dataset at hand and may not generalize well across different tasks. Other methods, such as Hou & Kung (2022); Tang et al. (2022); Wang et al. (2022), use sensitivity analysis and iterative rate search, which comes at a cost of increased computation overhead.

Our method offers a lightweight solution for automatically determining layer-wise retention rates. To that end, we first compute the Average Cosine Similarity (ACS) between patches at each layer (in an offline manner), which is used to capture the extent of layer-wise redundancy. This choice is motivated by previous research (Gong et al., 2021; Zhou et al., 2021) showing that deeper network layers tend to have higher patch similarity and, thus higher redundancy. Subsequently, we propose a rule for selecting the layer-wise retention rates that contain a single hyperparameter $\rho$. This features a *monotonicity* property in two aspects: a) the retention rate is decreasing with ACS (as motivated above), and b) the retention rate is increasing in $\rho$. It thus allows for efficient tuning of the proposed method to strike a salient balance between compression and accuracy. Additionally, we introduce a variant that leverages these rates as weighted percentile parameters for the layer-wise scores. This, in turn, gives rise to an adaptive rate selection method that is contingent on the input data.

In summary, our work introduces three key novelties: a) evaluation of inter-layer patch importances, b) fusion mechanism for integrating inter-layer and intra-layer importances, and c) a new auto-tuning mechanism for layer-wise retention rates with no overhead of re-training.

**Contributions:**
1) We propose SynergisTic pAtch pRuning (STAR) for Vision Transformers, which combines dynamic evaluation of patch importances within every single layer *during inference* together with *offline analysis* of patch importance across layers. Importance scores are fused by using a weighted exponentiation to determine the overall importance of each patch. STAR provides a more comprehensive view of the significance of each patch for the overall process and ensures that patches crucial for deeper layers are not mistakenly pruned based on shallower layer evaluation alone.
2) STAR incorporates an efficient means for automatically selecting the pruning rate at each layer. We propose a simple bilinear relation of the retention rate to a) the offline metrics (ACS) and b) a single control parameter $\rho$. This relationship establishes a crucial *monotonicity*, enabling effective

and efficient tunability of the proposed method to trade off compression vs. accuracy. Furthermore, we introduce an adaptive variant of the method, which yields input-dependent retention rates by utilizing the previous values as weighted percentile parameters with the same monotonicity property. 3) Through extensive experiments, we demonstrate superior performance in terms of both compression and accuracy over leading methods. Specifically, on the ImageNet dataset, our approach achieves higher accuracy with Top-1 Acc. degradation ranging from 0.2% to 0.3% while compressing the GFLOPs of DeiT-Base&Tiny by over 42% and 46%, respectively. The pruned DeiT-Small model achieves a throughput of 4,256 images/s, over 66% higher than the unpruned DeiT-Tiny model, while maintaining a substantial accuracy improvement of +6.8% Top-1 and +3.1% Top-5.

## 2 RELATED WORK

Research efforts have been dedicated to pruning Vision Transformer (ViT), aimed at improving inference efficiency and facilitating the deployment on devices with limited computational capabilities. Existing works on compressing ViTs fall into two main categories:

**Structure/Channel Pruning** focuses on eliminating redundant or less critical connections or weights to create a more compact and efficient network. While primarily developed for CNNs, methods based on sparse learning (Yu et al., 2022; Chen et al., 2021b) and sensitivity analysis (Yang et al., 2021) have been explored to reduce model redundancy in ViTs. Nevertheless, ViT divides input images into multiple patches, and computations are carried out in parallel across all patches. Pruning entire channels is static for all input, which may lead to a reduction in network capacity and limit the expressive power of the model (Chu et al., 2022; Hou & Kung, 2022). In response to this challenge, patch pruning has been proposed and it gives greater flexibility by allowing the dynamic selection of a subset of patches deemed useful for each layer.

**Patch Pruning** aims to remove redundant or less informative patches from the input image. This is based on the characteristic that ViT can process an arbitrary length of the input patch. It capitalizes on the self-attention mechanism in Transformer, which can thus effectively capture dependencies between patches in different layers (Peng et al., 2021). Tang et al. (2022) uses a top-down approach to compute importance scores for each patch by approximating its impact on the reconstruction error. Liang et al. (2022) introduce a fused patch representing the patches removed in each layer. Xu et al. (2022) keep patches with low-frequency components to convey more critical information to subsequent layers. However, all the aforementioned methods evaluate patch importance at a given layer and thus may fail to capture the cross-layer importance. Chen et al. (2023) automatically select compression rates by incorporating a sparsity-promoting regularizer into the loss controlled by a hyperparameter $\lambda$. Nevertheless, it requires retraining for different compression targets and does not address inter-layer dependencies explicitly.

To address this limitation, we propose a novel approach that incorporates inter-layer importance (inspired by model interpretability analysis) with intra-layer importance, offering a more comprehensive and direct view of the significance of each patch for the overall task. Besides, most methods rely on manually tuning the pruning rates at each layer (Yu et al., 2022; Yu & Wu, 2023; Chen et al., 2021b; Zhu et al., 2021). In contrast, STAR automates the pruning rate selection by pre-computing the ACS between patches within each ViT block.

## 3 ATTENTION MAP FOR DYNAMIC INTRA-LAYER PATCH EVALUATION

The attention operator at the $l$-th ($1 \le i \le L$) Transformer block for the $h$-th ($1 \le h \le H$) head is:

$$\text{Attention}(Q_l^h, K_l^h, V_l^h) = \text{Softmax}\left(\frac{Q_l^h K_l^{h\top}}{\sqrt{d_l^h}}\right) V_l^h, \tag{1}$$

where $Q_l^h, K_l^h, V_l^h$ are the query, key, and value of the $h$-th head in the $l$-th layer, $d_l^h = d/H$ with $d$ the input dimension and $H$ the heads number, and $\text{Softmax}\left(\frac{Q_l^h K_l^{h\top}}{\sqrt{d_l^h}}\right)$ is the *attention map*. Touvron et al. (2021) append the *class token* [CLS] to the tokens (patches) before the first layer. This traverses the ViT blocks and is used at the end to predict the class. The [CLS] plays the role of synthesizing information within a layer, and the attention map of it to the remaining tokens within a layer can be regarded as the contribution of each patch to the classification outcome. The [CLS] attention to other tokens in the $l$-th ViT block and $h$-th head can be written as (Vaswani et al., 2017):

$$\text{Attention}_{\text{[CLS]}}(q_{\text{class},l}^h, K_l^h, V_l^h) = \text{Softmax}\left(\frac{q_{\text{class},l}^h \cdot K_l^{h\top}}{\sqrt{d_l^h}}\right) V_l^h = attn_{(l)}^h \cdot V_l^h, \tag{2}$$

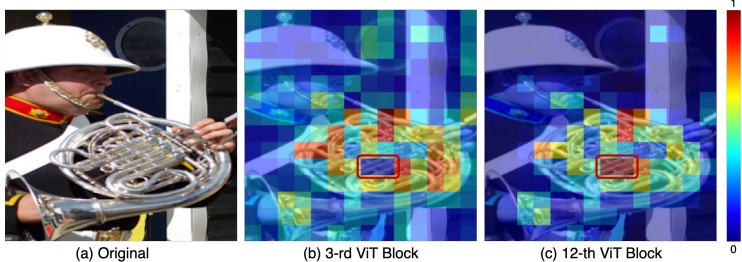

(a) Original       (b) 3-rd ViT Block      (c) 12-th ViT Block

Figure 1: Visualization of class attention in DeiT-Base using modified LRP for inter-layer patch importance. (a) is the original image; (b) and (c) are the visualization of the importance (plotted by a color code) of the image patches (tokens) at the third and twelfth Transformer layers (blocks), respectively. For example, the patch within the red square is considered unimportant (blue) for classification in the third layer but becomes important (red) in the twelfth layer. It is thus crucial to ensure that patches critical for deeper layers are not mistakenly pruned in shallower layers.

where $q^h_{\text{class},l}$ denotes the query vector of [CLS]. As mentioned above, the [CLS] of each layer aggregates the contributions of all image tokens for the classification task, as it can be viewed by treating $\text{Attention}_{[\text{CLS}]}$ as a linear combination of $V^h_l = [\boldsymbol{v}^h_1, \boldsymbol{v}^h_2, \dots, \boldsymbol{v}^h_N]$ value vectors, that correspond to the $N$ tokens. Therefore, the corresponding coefficient $attn^h_{(l,i)}$ can be considered as the importance of the $i$-th token at the $l$-th layer for the classification outcome for a given input, as in Liang et al. (2022); Wang et al. (2022). By collecting the results from all attention heads, the Intra-layer importance score in the $l$-th layer is obtained by averaging as: $\mathbf{S}^{\text{intra}}_{(l,i)} = \frac{1}{H} \sum_{h=1}^{H} attn^h_{(l,i)}$.

Note that $attn^h_{(l)}$ can be calculated in the forward process (i.e., sequentially, during inference), and the pruned patches are discarded from all subsequent Transformer layers; this directly reduces the computational costs involved (see Fig. 2a for an illustration). In our ablation study, the first part of Table 2 demonstrates that only using the Attention map $attn^h_{(l)}$ for patch pruning as in (Wang et al., 2022) can seriously affect the classification accuracy. The main reason for the model impairment is that pruning the output of each layer based on intra-layer information is rather aggressive. Tokens with low importance for [CLS] in the shallow blocks may be more important in the deeper blocks, which creates confusion in the pruning process because the intra-layer important scores only measure the importance for [CLS] *within a single layer*. Nevertheless, once patches are pruned, they cannot be inserted back. This motivates us to explore inter-layer dependencies to capture the overall importance of a patch in the whole model, which is discussed in the next section.

## 4 INTER-LAYER PATCH IMPORTANCE USING LRP

In this section, we describe how inter-layer patch importances can serve to improve compression. First, we present the key challenges and considerations that motivate us. Fig.1 illustrates the contribution of image patches to the final classification task at various layers within the DeiT-Base model, leveraging a modified LRP for inter-layer patch importance. This reflects the significance of each image patch to the [CLS] at the last layer and this visualization illustrates that the significance of a given patch varies across ViT blocks, which serves as a motivation for evaluating inter-layer importance throughout the entire classification task. For instance, patches with low importance for [CLS] in the shallow blocks (see Fig.1b) may become more important in the deeper blocks (see Fig.1c). Many existing methods do not fully exploit the redundancies present across different ViT blocks. This oversight can lead to essential patches required for deeper layers being mistakenly pruned based solely on evaluations from shallower layers. This serves as the motivation for combining dynamic evaluation of intra-layer patch importance scores within each individual layer during inference with offline calculations of inter-layer patch importance across layers. This combined approach provides a more comprehensive assessment of patch importance within ViTs.

In the following, we will recap LRP first, and then the modifications we made on it. Details of the modified LRP are described in Appendix B. The Layer-wise Relevance Propagation (LRP) method, has proven to be effective in interpreting neural networks (Bach et al., 2015). A variant of LRP has been specifically designed for Vision Transformer (Chefer et al., 2021) using the GELU activation (Hendrycks & Gimpel, 2016). This analysis can give both positive and negative values. We only consider the relevance scores that have a positive value as described by Chefer et al. (2021). In our modification, a) instead of operating at the pixel level, we carry an additional aggregation of

multiple pixels' relevance belonging to the same patch and b) rather than focusing solely on capturing the relationship between one pixel and one class, we have adapted the original LRP to capture the correlation between one patch and multiple classes.

The calculation of *inter-layer patch importance* $\mathbf{S}_{(l)}^{\text{inter}}$ is carried out in an offline fashion on the training dataset (with no overhead to the inference). It serves to assess the importance of each patch from the viewpoint of the overall classification process and obtain *statistical information* based on the dataset. During inference, these pre-competed inter-layer patch importances are combined with intra-layer importances, which are obtained dynamically based on the input for the [CLS] attention map in each layer to achieve complementarity. In the subsequent section, we delineate our approach for fusing the intra-layer and inter-layer importance scores.

## 5 SYNERGISTIC PATCH PRUNING (STAR) FOR VITS

Intra-layer analysis derives the importance score of each image patch for the class token [CLS] within a layer based on the linear relationship between the attention map and the value matrix in (2). However, the linear relationship is valid *within a single layer*, and not for the whole model (due to the Softmax operation). Thus, intra-layer scores cannot fully capture the patch importance for the overall classification task. In contrast, the inter-layer analysis by LRP reflects the contribution of each image token to the [CLS] at the last layer. In summary, the intra-layer and inter-layer importance information can complement each other and synthesize more comprehensive criteria for patch pruning. The inter-layer method can compensate for the lack of inter-layer information in the intra-layer method by pre-computing scores offline using Alg. 2 in Appendix B. The intra-layer method operates sequentially in an online manner (see steps 5,6 of Alg. 1), and so does our proposed method, STAR.

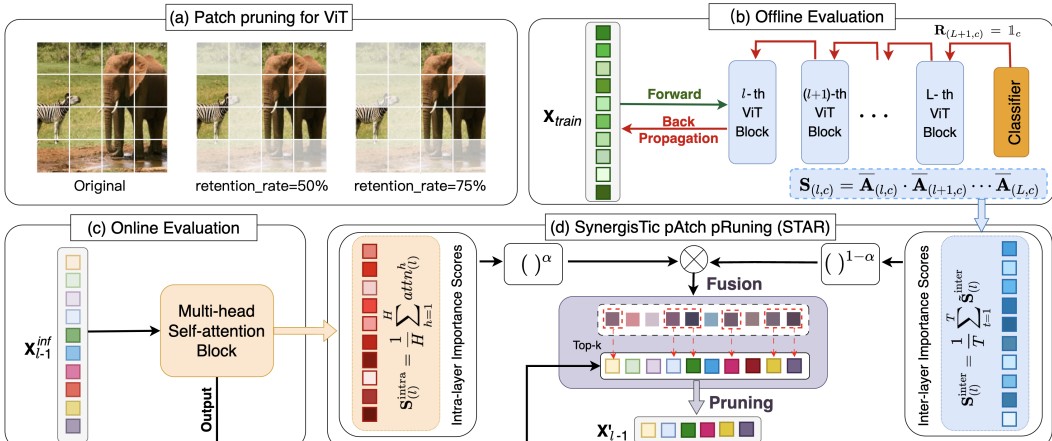

Figure 2: (a) Visualization of patch pruning for ViT (according to the listed retention rates, from the original 16 patches, 8 patches are kept in the second layer, and 6 at the third). (d) Overview of SynergisTic pAtch pRuning (STAR). Inter-layer importance scores (b) for each layer are computed offline using the LRP method during back-propagation on the training dataset. Intra-layer importance scores (c) are obtained online (i.e., during inference) using the attention map of each layer's [CLS]. The inter-layer importance score $\mathbf{S}_{(l,i)}^{\text{inter}}$ and the intra-layer importance score $\mathbf{S}_{(l,i)}^{\text{intra}}$ corresponding to the $i$-th token at the $l$-th block are fused as $\mathbf{S}_{(l,i)}^{\text{intra}}{}^{\alpha} \times \mathbf{S}_{(l,i)}^{\text{inter}}{}^{(1-\alpha)}$. Pruning is conducted by retaining the top-$k$ important patches based on the fused scores.

### 5.1 FUSION OF INTRA-/INTER-LAYER IMPORTANCE SCORES

We opt to carry fusion of the intra-layer and inter-layer scores (see Fig. 2d and Alg. 1), so as to capture the impact of patch pruning on a single-layer basis as well as incorporate inter-layer relations pertaining to the whole classification task. First, note that both scores are positive (due to the Softmax operation in (2) and the restriction to positive values in (8)). Therefore, after normalization, they can be viewed as probability mass functions (pmfs). For simplicity, we denote by $p^1$ the pmf corresponding to intra-layer scores and by $p^2$ the pmf for inter-layer scores. The Kullback–Leibler (KL) divergence is a widely used distance for pmfs, that is defined as $KL(p,q) := \sum_i p_i \log \frac{p_i}{q_i}$, for pmfs $p, q$ with common support. Our fusion method amounts to calculating a weighted centroid

(with parameter $\alpha \in [0,1]$) with respect to the KL divergence using the following formulation:

$$\underset{p}{\text{minimize}} \quad \alpha KL\left(p\|p^1\right) + (1-\alpha)KL\left(p\|p^2\right), \quad s.t. \quad p_i \geq 0, \; \sum_i p_i = 1, \tag{3}$$

where $\alpha$ is a hyperparameter that balances the contribution of intra-/inter-layer scores. This problem admits a closed-form solution:

$$p_i = \frac{\left(p_i^1\right)^\alpha \cdot \left(p_i^2\right)^{1-\alpha}}{\sum_i \left(p_i^1\right)^\alpha \cdot \left(p_i^2\right)^{1-\alpha}}, \tag{4}$$

that is $S^{\text{fused}} \propto S^{\text{intra} \, \alpha} \times S^{\text{inter} \, (1-\alpha)}$ (the derivation is provided in Appendix C). Given the retention rates, layer-wise pruning is carried by maintaining the top-$k$ important patches in the $l$-th layer (step 9 of Alg. 1; note that [CLS] is excluded from this operation), sequentially. In the following, we provide a method to automatically select retention rates at each layer based on metrics that are obtained offline (i.e., as a pre-processing step).

---

**Algorithm 1** SynergisTic pAtch pRuning (STAR)

---

**Input**: Training dataset $\mathcal{D}$, test dataset $\mathcal{E}$, Vision Transformer $\mathcal{T}$ with $L$ layers, hyperparameter $\alpha$ for fusion, average cosine similarity ACS (pre-computed offline from the training data), hyperparameter $\rho$ controlling the FLOPs Drop.

1: $\mathbf{S}^{\text{inter}} \leftarrow \text{InterIS}\left(\mathcal{T}, \mathcal{D}, \gamma, T\right)$ [Alg. 2].
2: **for** $e$ in $\mathcal{E}$ **do**
3:     Initialize the input of the first block $X_1 = e$.
4:     **for** $l = 1, \ldots, L$ **do**
5:         Calculate $attn_{(l)}^h = \text{Softmax}\left(\frac{q_{\text{class},l}^h \cdot K_l^{h\top}}{\sqrt{d_l^h}}\right)$.
6:         Calculate $\mathbf{S}_{(l)}^{\text{intra}} = \frac{1}{H}\sum_{h=1}^H attn_{(l)}^h$.
7:         Calculate $\mathbf{S}_{(l)}^{\text{fused}} = \mathbf{S}_{(l)}^{\text{intra} \, \alpha} \times \mathbf{S}_{(l)}^{\text{inter} \, (1-\alpha)}$.
8:         Set $p_l = \frac{\rho-1}{\text{ACS}_L - \text{ACS}_1}\text{ACS}_l + \frac{\text{ACS}_L - \rho \, \text{ACS}_1}{\text{ACS}_L - \text{ACS}_1}$.
9:         $\Bigg\{$  **Option 1:** *Static retention rate selection*
                Keep $p_l\%$ patches with top-$k$ ($\mathbf{S}_{(l)}^{\text{fused}}$) scores.
            **Option 2:** *Adaptive retention rate selection*
                $\tilde{\mathbf{S}}_{(l)}^{\text{fused}} \leftarrow$ Normalize and sort $\mathbf{S}_{(l)}^{\text{fused}}$.
                Calculate $r_l = \text{idx}\left(\text{CDF}(\tilde{\mathbf{S}}_{(l)}^{\text{fused}}) \geq p_l\right)$.
                Keep $r_l$ patches with top-$k$ ($\tilde{\mathbf{S}}_{(l)}^{\text{fused}}$) scores.
10:     **end for**
11: **end for**

**Output**: The pruned Vision Transformer $\mathcal{T}'$.

---

## 5.2   Design of the retention rates

Fig. 1 shows the visualization of the contribution of the image patches to the final classification task at different layers: Fig. 1c illustrates that the attention scores of image patches to [CLS] usually have a clustering effect at a larger depth of the network, while they are relatively scattered in a shallow layer (Fig. 1b). This indicates that as the network gets deeper, there is an increasing number of patches that can be pruned due to their minimal impact on the final classification task. In our proposed method, each layer uses a different retention rate $p_l$ (thus, the number of retained patches up until layer $l$ is the running product $p_1 \cdots p_l$). The rise in cosine similarity between image patches as the network goes deeper (Zhou et al., 2021) motivates us to employ the average cosine similarity (ACS) calculation among the image patches $X_l \in \mathbb{R}^{N \times d}$ produced by each ViT block to automatically determine the retention rates $p_l$. Specifically, ACS is calculated as follows:

$$\text{ACS}_l = \frac{2}{N(N+1)}\sum_{i=0}^N \sum_{j=i+1}^N \frac{X_l^i \cdot X_l^j}{||X_l^i||_2 ||X_l^j||_2}. \tag{5}$$

Based on this, we establish a linear relationship between the retention rate $p_l$ of each layer and the ACS, which can be used for *static retention rate selection* (**Option 1** in Step 9 of Alg. 1; static means that these rates are the same for any input to the model), i.e., $p_l = a\,\text{ACS}_l + b$. We select the parameters $(a, b)$ based on two principles: 1) since the first layer is the most sensitive to pruning, we retain all patches (i.e. $p_1 = 1$) and 2) since the last layer is typically less sensitive, we set $p_L = \rho$, where $\rho \in (0, 1)$ can be used as the single control parameter to steer the desired FLOPs Drop. By

solving the resulting $2 \times 2$ linear system: $1 = p_1 = a\,\text{ACS}_1 + b, \rho = p_L = a\,\text{ACS}_L + b$, we obtain the expression in Step 8 of Alg. 1. Note that this is bilinear in $(\text{ACS}_l, \rho)$ (i.e., it is linear to each one for the other fixed). Given that $\text{ACS}_1$ is the smallest value across $l = 1, \ldots, L$ (this is verified experimentally), we can deduce a monotonicity property in that for all $l$: a) $p_l$ decreases with $\text{ACS}_l$ (since $\rho < 1$) and b) $p_l$ increases with $\rho$, for all $l \neq 1$ (since $\text{ACS}_l > \text{ACS}_1$). This is a desirable trait, as it allows for efficiently tuning the method to achieve a target overall FLOPs reduction (e.g., using bisection), in order to trade off compression vs. accuracy.

Additionally, we introduce an alternate *adaptive retention rate selection*, which we call a-STAR (**Option 2** in Step 9 of Alg. 1; adaptive means that the retention rate depends on the input). This is done by using the calculated value $p_l$ as a weighted percentile parameter for the normalized fused importance scores (CDF stands for cumulative distribution function). The same monotonicity property holds for this variant and additionally, we prove that its compression is no less than that of Option 1 (see Appendix D).

## 6 EXPERIMENTS

STAR is evaluated on the benchmark ImageNet (ILSVRC2012) dataset using two popular ViT models, namely DeiT (Touvron et al., 2021) (three different sizes: DeiT-Tiny, DeiT-Small, DeiT-Base; the code is modified based on the study of Meta Research), and LV-ViT (Jiang et al., 2021) (two different sizes: LV-ViT-S, LV-ViT-M). All experiments are deployed with PyTorch on NVIDIA A100 & 3090 GPUs. Instead of training the network from scratch, we directly use the pre-trained models. The pruned models are fine-tuned for 120 epochs with hard distillation (Touvron et al., 2021) of their corresponding original models. All throughputs are measured during inference on an A100 GPU with a batch size of 128 and FP32. The learning rate is reduced from 2e-5 to 2e-9 with a cosine schedule. We refer the reader to Appendix E for the experimental setting details. Our code is available here.

### 6.1 EXPERIMENTAL RESULTS

We compare our proposed method with several representative model compression methods, including SCOP (a technique inspired by CNN channel pruning) (Tang et al., 2020), PoWER (Goyal et al., 2020), VTP (Zhu et al., 2021), HVT (Pan et al., 2021b), SAViT (Zheng et al., 2022), IA-RED[2](Pan et al., 2021a), EViT (Liang et al., 2022), and Patch Slimming (PS) (Tang et al., 2022); multi-dimensional pruning CP-ViT (Song et al., 2022), and VTC-LFC (Wang et al., 2022); token merging pruning ToMe (Bolya et al., 2023). Through extensive experimentation, it is portrayed that STAR achieves formidable compression while also maintaining high model accuracy, as shown in Table 1. For full results, see Appendix E.

**Results on DeiTs:** For the DeiT-Base model, STAR achieves a substantial reduction of 42.0% in FLOPs, with a negligible drop of only 0.3% in Top-1 Acc. and 0.1% in Top-5 Acc. Even at a higher level of FLOPs reduction, namely 47.2% (+4.0% over VTP), the drop in Top-5 Acc. is only 0.5%, and the drop in Top-1 Acc. is 0.8%. We also assessed the acceleration in inference speed using a single A100 GPU with a batch size of 128. The DeiT-Base model under the two mentioned pruning levels achieved a speedup of 69.5% and 84.9% after pruning.

The DeiT-Small model originally has a much lower computational cost of 4.6 GFLOPs than DeiT-Base but still greatly benefits from pruning with STAR. STAR achieves a reduction of 43.5%/50% with corresponding speedups of 83.9% and 104.5% on a single A100 GPU, with minimal accuracy drop (0.4%/0.6% for Top-1 and 0.5%/0.7% for Top-5). These speedups surpass other methods, and so do the respective accuracies. The merits are further accentuated by measuring the achieved throughput (the number of images that can be processed per second). Remarkably, the pruned DeiT-Small reaches a throughput of 4,080.6 images/s, which is over 60% higher than the much smaller (unpruned) DeiT-Tiny model, while having a substantially higher accuracy (+7.0% Top-1 and +3.2% Top-5). Even for the smallest-scale DeiT-Tiny model, our method showcases a twofold increase in throughput by reducing the computational cost to just 0.7 GFLOPs. Despite this significant reduction in computation, the degradation in Top-1 Acc. is minimal, at only 0.2%. Compared to VTC-LFC, it exhibits a +27.1% speedup and a 1.0% improvement in Top-1 Acc. Last but not least, even for cases where our method has the same FLOPs reduction as another method (i.e., SCOP on DeiT-Base/-Small), in addition to superior accuracy, it achieves much higher throughput because of its lightweight nature as an online pruning method.

**Results on LV-ViTs:** We conducted experiments using two LV-ViT models. For the medium-sized LV-ViT-M model, STAR achieved a Top-1 Acc. of 83.3% and a Top-5 Acc. of 96.4%. This is an

Table 1: Experimental results of SynergisTic pAtch pRuning. 'STAR' means using the static retention rate selection (Option 1 in Step 9 of Alg. 2) while 'a-STAR' refers to using the adaptive retention rate selection (Option 2). The hyperparameter $\rho$ (Alg. 1) controls the compression rate. 'FLOPs $\downarrow$' denotes the FLOPs reduction ratio, while the throughput measures the number of images that can be processed per second (including the cost of both online patch pruning and inference). The speedup (i.e., increase of throughput) is obtained on an A100 GPU. The marker * indicates the result is from the corresponding articles. The best results in each column are depicted in boldface.

| Model | Method | $\rho$ | Top-1 Acc. | Top-5 Acc. | GFLOPs | FLOPs↓ (%) | Throughput | Speedup (%) |
|---|---|---|---|---|---|---|---|---|
| | Baseline | - | 81.8 | 95.6 | 17.6 | - | 1,051.6 | - |
| | IA-RED² | - | 80.3 | 95.0 | 11.8 | 33.0 | 1,380.1 | 31.2 |
| | EViT | - | 81.3 | 95.3 | 11.6 | 34.1 | 1,454.7 | 38.3 |
| | CP-ViT | - | 81.1 | - | 10.3 | 41.6 | - | - |
| DeiT-Base | SCOP | - | 79.7 | 94.5 | 10.2 | 42.0 | 1,504.7 | 43.1 |
| | **STAR (Ours)** | 0.75 | **81.5** | **95.5** | 10.2 | 42.0 | 1,782.8 | 69.5 |
| | **a-STAR (Ours)** | 0.75 | 81.4 | **95.5** | **10.0** | **43.2** | **1,867.0** | **77.5** |
| | VTP | - | 80.7 | 95.0 | 10.0 | 43.2 | 1,586.0 | 50.8 |
| | **STAR (Ours)** | 0.7 | **81.0** | **95.1** | 9.3 | 47.2 | 1,944.5 | 84.9 |
| | **a-STAR (Ours)** | 0.7 | 80.8 | 95.0 | **9.0** | **48.9** | **2,028.2** | **92.8** |
| | Baseline | - | 79.8 | 95.0 | 4.6 | - | 1,995.6 | - |
| | IA-RED² | - | 79.1 | **94.5** | 3.2 | 31.5 | 2,566.4 | 28.6 |
| | SCOP | - | 77.5 | 93.5 | 2.6 | 43.5 | 2,823.8 | 41.5 |
| | PoWER | - | 78.3 | 94.0 | 2.7 | 41.3 | 2,939.5 | 47.3 |
| DeiT-Small | ToMe * | - | **79.4** | - | 2.7 | 41.3 | 3,346.7 | 67.7 |
| | **STAR (Ours)** | 0.75 | **79.4** | **94.5** | 2.6 | 43.5 | 3,670.8 | 83.9 |
| | **a-STAR (Ours)** | 0.75 | 79.3 | 94.4 | **2.5** | **45.7** | **3,843.7** | **92.6** |
| | HVT | - | 78.0 | 93.8 | 2.4 | 47.8 | 2,835.7 | 42.1 |
| | **STAR (Ours)** | 0.7 | **79.2** | **94.3** | 2.3 | 50.0 | 4,080.6 | 104.5 |
| | **a-STAR (Ours)** | 0.7 | 79.0 | 94.2 | **2.2** | **52.2** | **4,256.9** | **113.3** |
| | Baseline | - | 72.2 | 91.1 | 1.3 | - | 2,544.0 | - |
| | SAViT | - | 70.7 | - | 0.9 | 30.8 | - | - |
| DeiT-Tiny | ToMe * | - | 71.4 | - | 0.8 | 41.7 | 3,137.1 | 57.2 |
| | VTC-LFC | - | 71.0 | 90.4 | 0.8 | 41.7 | 4,352.8 | 71.1 |
| | **STAR (Ours)** | 0.8 | **72.0** | **90.7** | **0.7** | **46.2** | 5,042.9 | 98.2 |
| | **a-STAR (Ours)** | 0.8 | **72.0** | **90.7** | **0.7** | **46.2** | **5,104.8** | **99.8** |
| | Baseline | - | 83.8 | 96.6 | 12.6 | - | 1,061.6 | - |
| LV-ViT-M | EViT | - | 82.9 | 95.8 | 8.5 | 32.5 | 1,820.6 | 71.5 |
| | **STAR (Ours)** | 0.8 | **83.3** | **96.4** | 6.7 | 46.8 | 1,922.6 | 81.1 |
| | **a-STAR (Ours)** | 0.8 | 83.2 | 96.3 | **6.5** | **48.4** | **1,971.7** | **85.7** |
| | Baseline | - | 83.3 | 96.3 | 6.5 | - | 1,681.3 | - |
| | PS-LV-ViT-S * | - | 82.4 | - | 4.7 | 27.7 | - | - |
| LV-ViT-S | EViT | - | 82.5 | **96.2** | 3.9 | 40.0 | 2,374.0 | 41.2 |
| | VTC-LFC | - | 81.8 | 95.6 | 3.2 | 50.8 | - | - |
| | **STAR (Ours)** | 0.65 | **82.7** | **96.2** | 3.2 | 50.8 | 3,353.2 | 99.4 |
| | **a-STAR (Ours)** | 0.65 | 82.5 | 96.1 | **3.1** | **52.3** | **3,439.8** | **104.6** |

increase of +0.4%/+0.6% over EViT, while STAR also achieves +9.6% higher speedup. Similarly, the experimental results for LV-ViT-S outperform other approaches across all evaluation metrics, achieving a high acceleration of 99.4%.

**Additional results of a-STAR:** Compared to STAR, a-STAR adaptively adjusts the retention rate for different inputs. Notably, a-STAR achieves higher compression and throughput, while STAR demonstrates better accuracy when using the same $\rho$. The only overhead of a-STAR is the additional calculation of CDF after sorting, which is negligible. For instance, in the case of DeiT-Small with $\rho = 0.75$, a-STAR exhibits a +2.2% increase in flops drop and +8.7% in throughput compared to STAR. However, STAR maintains a +0.1% higher Top-1 and Top-5 accuracy than a-STAR.

## 6.2 ABLATION STUDY

To further analyze the impact of different components on the proposed method, we conducted separate experiments to test the influence of the intra-layer and inter-layer scoring mechanisms, using only intra-layer scores together with our automatic pruning schedules based on ACS, as well as the effectiveness of KL fusion. All results reported in this section are obtained before fine-tuning.

**Effectiveness of intra-layer & inter-layer scores:** For the former two parts in Table 2, the retention rates at each layer were derived empirically using a product rule. A significant decrease in classification accuracy is observed for using intra-layer scores only for patch pruning. With a GFLOPs reduction to 10.2 (as indicated in the last row of the Intra-layer pruning part in the table), the Top-1 Acc. drops by 2.4% and the Top-5 Acc. drops by 1.4%. Furthermore, when aiming for higher compression rates with retention rates of 0.5, $0.25 = 0.5^2$, and $0.125 = 0.5^3$ only at layers

Table 2: Ablation for the effectiveness of different modules in STAR on the ImageNet dataset for DeiT-Base. Intra-/Inter-layer pruning refers to using only intra-/inter-layer importance scores in the top-$k$ selection for pruning. Furthermore, the pruning schedules are designed manually: the prune_layers indicate the selected layers for patch pruning (no pruning is carried in other layers), and retention_rate represents the patch keep ratio for each selected layer. Top-1 Acc. and Top-5 Acc. reported in the table reflect the accuracy without fine-tuning. An Inter-layer pruning sample of $T = 10$ rounds is used to get the inter-layer information. ACS method refers to automatically determining the layer-wise retention rate using the mechanism presented in Sec. 5.2 and only intra-layer importance, while STAR and a-STAR additionally use the fusion.

| Method | $\rho$ | prune_layers | retention_rate | batch_size | Top-1 Acc. | Top-5 Acc. | GFLOPs |
|---|---|---|---|---|---|---|---|
| DeiT-Base | - | - | - | - | 81.8 | 95.6 | 17.6 |
| | - | [4,7,10] | $0.5 \& 0.5^2 \& 0.5^3$ | - | 71.1 | 88.4 | 7.8 |
| Intra-layer pruning | - | [5,8,10] | $0.7 \& 0.7^2 \& 0.7^3$ | - | 78.0 | 93.1 | 11.0 |
| | - | **[4,7,10]** | **$0.7 \& 0.7^2 \& 0.7^3$** | - | **79.4** | **94.2** | **10.2** |
| | - | [5,8,10] | $0.7 \& 0.7^2 \& 0.7^3$ | 64 | 64.1 | 79.6 | 11.0 |
| Inter-layer Pruning | - | [5,8,10] | $0.70 \& 0.7^2 \& 0.7^3$ | 128 | 77.3 | 93.1 | 11.0 |
| | - | **[4,7,10]** | **$0.7 \& 0.7^2 \& 0.7^3$** | 128 | **78.2** | **93.9** | **10.2** |
| | 0.65 | - | - | - | 76.6 | 92.6 | 8.4 |
| ACS | 0.7 | - | - | - | 78.2 | 93.6 | 9.3 |
| | 0.75 | - | - | - | **79.7** | **94.5** | **10.2** |
| **STAR (Ours)** | 0.75 | - | - | 128 | 80.0 | 94.7 | **10.2** |
| **a-STAR (Ours)** | 0.77 | - | - | 128 | **80.2** | **94.8** | **10.2** |

4, 7, and 10, respectively, the Top-1 Acc. drops by an unacceptable 10.7% compared to the un-pruned model, and the Top-5 Acc. decreases by 7.2%. This illustrates that intra-layer scores alone are ineffective, as they fail to protect those patches critical for deeper layers from being pruned at a shallower layer, as explained in Sec. 4. Besides, pruning based solely on inter-layer information is not sufficient either. For instance, when the GFLOPs drop to 10.2, the Top-1 Acc. decreases by 3.6% and Top-5 Acc. decreases by 1.7% compared to the original model. This is because this approach alone lacks any adaptability to the input (since the scores are obtained through offline aggregation). These findings serve in support of our choice to fuse the scores.

**Influence of automatic pruning schedules based on ACS:** The ACS module is used to automatically determine the layer-wise retention rate based on the target compression rate ($\rho$) at the last layer, as explained in Sec. 5.2. In Table 2, we used this method without fusion (i.e., only on intra-layer scores). Compared to the manually chosen pruning schedules (Intra-layer pruning part of Table 2), the ACS-based generation achieves better performance for comparable FLOPs reduction. For instance, by setting $\rho = 0.75$, ACS achieves the same GFLOPs (10.2) with improvement (+0.3%) in both Top-1 and Top-5 Accuracy. We sampled 500 images to calculate the ACS between patches in our experiment. Details about using different data sizes to calculate ACS are listed in Table 6, Appendix E.

**Effectiveness of KL fusion:** The last part of Table 2 employs the ACS module on the fusion result of intra-/inter-layer importance scores. We present two variants: STAR and a-STAR, corresponding to the two options of static and dynamic retention rate selection as outlined in Alg. 1 (Step 9). For the static ratio setting (STAR), our results demonstrate improvements in both Top-1 Acc. (+0.6%) and Top-5 Acc. (+0.5%) compared to using intra-layer importance scores alone. With the same GFLOPs (10.2), the adaptive pruning strategy (a-STAR) achieves a +0.2%/+0.1% additional improvement on Top-1 and Top-5 Acc. over STAR. These findings underscore the effectiveness of fusing intra-/inter-layer scores in the context of patch selection. In our experiments, we use $\alpha = 0.15$ in DeiT models and $0.3$ for LV-ViT models. Details of the selection of the hyperparameter $\alpha$ in the KL fusion module of STAR are listed in Table 3, Appendix E.

## 7 CONCLUSION

STAR is a dynamic patch pruning method tailored for Vision Transformers (ViTs). It combines intra-layer importance scores, which capture the importance of patches at a specific layer, with inter-layer importance scores that consider the variable impact of patches across layers on the final classification outcome. By automatically designing pruning schedules based on these fused importance scores, STAR substantially reduces computational costs while maintaining high accuracy. Numerous comparative experiments on the DeiT and LV-ViT models illustrated improvements in terms of both higher compression and accuracy. Besides, STAR is lightweight as it can be unraveled from the consistently higher throughput it achieves (even for common FLOPs reduction).

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

## APPENDIX

This appendix is composed of five parts. Section A presents a detailed description of the Vision Transformer (ViT) architecture. The algorithm of using (a slight modification of) LRP to calculate the inter-layer patch importance is described in Appendix B. The details about the derivation of the fusion mechanism are listed in Section C, while the adaptive retention rate selection is discussed in Section D. Finally, the full experimental results are presented in Section E.

## A  PRELIMINARIES

A Vision Transformer (ViT) block includes a multi-head self-attention (MSA) block and a multi-layer perceptron (MLP) block. The input image $X_0^{(1)} \in \mathbb{R}^{D \times W \times C}$ of size $D \times W$ and channel number $C$ is reshaped into flattened patches $X_0^{(2)} \in \mathbb{R}^{N \times (P^2 C)}$: the image is split into $N = (D \cdot W)/P^2$ patches of size $P \times P$ that are treated the same way as tokens (words) in an NLP application. The patches are then projected into a $d$-dimensional embedding space to obtain $X_0^{(3)} \in \mathbb{R}^{N \times d}$. Besides, (Dosovitskiy et al., 2021) proposed to add a learnable embedding to the sequence of patch embeddings as a *class token* [CLS]. Notice that since the Transformer block does not limit the size of input $N$, it is always possible to add a class token before data processing for Transformer models that do not inherently have [CLS]. The [CLS] serves to aggregate patch information and is directly responsible for the classification task, thus the final input that includes [CLS] is given by $X_0 \in \mathbb{R}^{(N+1) \times d}$. At the $l$-th ($1 \leq i \leq L$) block, the MSA and MLP modules operate as follows:

$$X_l' = \text{MSA}(X_{l-1}) + X_{l-1} = \text{Concat}\left[\text{Softmax}\left(\frac{Q_l^h K_l^{h\top}}{\sqrt{d_l^h}}\right) V_l^h\right]_{h=1}^{H} W_l^o + X_{l-1},$$

$$X_l = \text{MLP}(X_l') = \phi\left(X_l' W_l^1\right) W_l^2 + X_l', \tag{6}$$

where $Q_l^h = X_{l-1} W_l^{h,q}$, $K_l^h = X_{l-1} W_l^{h,k}$, and $V_l^h = X_{l-1} W_l^{h,v}$ are the query, key, and value of the $h$-th head in the $l$-th layer. Note that $X_l \in \mathbb{R}^{(N+1) \times d}$ for all $l = 1, \ldots, L$ and the dimensions of the weight matrices are given by: $W_l^{h,q} \in \mathbb{R}^{d \times d_q}, W_l^{h,k} \in \mathbb{R}^{d \times d_k}, W_l^{h,v} \in \mathbb{R}^{d \times d_v}$, $d_q = d_k = d_v = d_l^h = d/H$, where $H$ is the number of heads. After (column-wise) concatenation, a learnable projection $W_l^o \in \mathbb{R}^{d \times d}$ is applied to the multi-head structure. $W_l^1$ and $W_l^2$ are the weights for the MLP block and $\phi$ represents the activation function (e.g., GELU) that is applied element-wise. The above description shows that the ViT block can operate for an arbitrary number of input patches. Thus, it is feasible and reasonable to reduce the amount of model computation by reducing the number of patches, a concept typically referred to as *patch pruning*.

## B  MODIFIED LRP FOR INTER-LAYER PATCH IMPORTANCE

In this section, we present more details on the modified LRP for capturing inter-layer patch importance. The relevance propagation in LRP follows the Deep Taylor Decomposition (Montavon et al., 2017):

$$R_l^j = \mathcal{G}(\mathbf{X}_l, \mathbf{W}_l, \mathbf{R}_{l+1}) = \sum_i x_l^j \frac{\partial L_l(\mathbf{X}_l, \mathbf{W}_l)}{\partial x_l^j} \frac{R_{l+1}^i}{L_l(\mathbf{X}_l, \mathbf{W}_l)}, \tag{7}$$

where $l \in \{1, \ldots, L\}$ is the layer index in a network and $L_l(\mathbf{X}_l, \mathbf{W}_l)$ is the $l$-th layer's operation on the input feature map $\mathbf{X}_l$ with weight $\mathbf{W}_l$; $x_l^j$ refers to the $j$-th element of the input and $R_l^j$ is the $j$-th relevance value in layer $l$. For Vision Transformer using the GELU activation, this analysis can give both positive and negative values. We only consider the relevance scores that have a positive value as described by Chefer et al. (2021) to avoid the mix between positive and negative values which may lead to an inaccurate high relevance score. In our modification, instead of operating at the pixel level, we carry an additional aggregation of multiple pixels' relevance belonging to the same patch. Relevance $R_{(l,c)}^j$ of the $j$-th patch to the target class $c \in$ top-$\gamma$ classes $\mathcal{C}$ on layer $l$ is calculated as follows (Chefer et al., 2021):

---

**Algorithm 2** Inter-layer importance scores using LRP.

---

**Input**: Vision Transformer $\mathcal{T}$ with $L$ layers, training dataset $\mathcal{D}$, number of highest-Acc. classes $\gamma$, number of rounds $T$.

1: **for** $t = 1, \ldots, T$ **do**
2:     Randomly sample a batch $\mathcal{B}$ from $\mathcal{D}$.
3:     **for** $b$ in $\mathcal{B}$ **do**
4:         Retrieve the classes $\mathcal{C}$ with top-$\gamma$ highest Acc.
5:         **for** $c$ in $\mathcal{C}$ **do**
6:             Initialize $\mathbf{R}_{(L+1,c)}$ as $\mathbb{1}_c$ (one-hot encoding of the target class $c$).
7:             Compute the gradients of all layers $\nabla \mathbf{A}$;
8:             **for** $l = L, \ldots, 1$ **do**
9:                 $R^j_{(l,c)} = \sum_{\{i|(i,j)\in Q\}} \frac{x^j_l w^{ji}_l}{\sum_{\{j'|(j',i)\in Q\}} x^{j'}_l w^{j'i}_l} R^i_{(l+1,c)}$,   // where $Q := \{(i,j) \mid x^j_l w^{ji}_l \geq 0\}$
10:                 $\overline{\mathbf{A}}_{(l,c)} = I + \mathbb{E}_h \left( \nabla \mathbf{A}^h_l \odot \mathbf{R}_{(l,c)} \right)^+$,
11:                 $\mathbf{S}_{(l,c)} = \overline{\mathbf{A}}_{(l,c)} \cdot \overline{\mathbf{A}}_{(l+1,c)} \cdots \overline{\mathbf{A}}_{(L,c)}$,
12:                 $\overline{\mathbf{S}}^{\text{inter}}_{(l,c)} = \mathbf{S}_{(l,c)}[1][:]$.
13:             **end for**
14:         **end for**
15:     **end for**
16:     **for** $l = 1, \ldots, L$ **do**
17:         $\tilde{\mathbf{S}}^{\text{inter}}_{(l)} = \frac{1}{|\mathcal{B}| \cdot |\mathcal{C}|} \sum_{b \in \mathcal{B}} \sum_{c \in \mathcal{C}} \overline{\mathbf{S}}^{\text{inter}}_{(l,c)}$.
18:     **end for**
19: **end for**
20: **for** $l = 1, \ldots, L$ **do**
21:     $\mathbf{S}^{\text{inter}}_{(l)} = \frac{1}{T} \sum^T_{t=1} \tilde{\mathbf{S}}^{\text{inter}}_{(l)}$.
22: **end for**

**Output**: The inter-layer importance scores $\mathbf{S}^{\text{inter}}$.

---

$$R^j_{(l,c)} = \mathcal{G}_Q \left( \mathbf{X}_l, \mathbf{W}_l, \mathbf{R}_{(l+1,c)} \right) = \sum_{\{i|(i,j)\in Q\}} \frac{x^j_l w^{ji}_l}{\sum_{\{j'|(i,j')\in Q\}} x^{j'}_l w^{j'i}_l} R^i_{(l+1,c)}, \tag{8}$$

where $l$ denotes the ViT block index, $Q := \{(i,j) \mid x^j_l w^{ji}_l \geq 0\}$ is the subset of patch index pairs corresponding to positive relevance values and indices $i, j$ represent image patches. $\mathbf{X}_l$ and $\mathbf{W}_l$ denote the input and weights (the compound of $W^{h,q}_l \in \mathbb{R}^{d \times d_q}, W^{h,k}_l \in \mathbb{R}^{d \times d_k}, W^{h,v}_l \in \mathbb{R}^{d \times d_v}$ for the $h$-th attention head) of layer $l$. To initialize the relevance propagation, we set the patch relevance $\mathbf{R}_{(L+1,c)} = \mathbb{1}_c$ as a one-hot encoding for the target classes $c$.

Following the propagation procedure of relevance and gradients and denoting each attention map by $\mathbf{A}_l \in \mathbb{R}^{H \times (N+1) \times (N+1)}$ for block $l$, the relevance scores $\mathbf{R}_{(l,c)}$ along with the gradients $\nabla \mathbf{A}_l$ are used to recursively compute the inter-layer patch importance scores $\mathbf{S}_{(l,c)} \in \mathbb{R}^{(N+1) \times (N+1)}$ based on:

$$\mathbf{S}_{(l,c)} = \overline{\mathbf{A}}_{(l,c)} \cdot \overline{\mathbf{A}}_{(l+1,c)} \cdots \overline{\mathbf{A}}_{(L,c)}, \text{ where } \overline{\mathbf{A}}_{(l,c)} = I + \mathbb{E}_h \left( \nabla \mathbf{A}^h_l \odot \mathbf{R}_{(l,c)} \right)^+, \tag{9}$$

and $\odot$ is the Hadamard product, $\mathbb{E}_h$ denotes averaging across the 'heads' dimension, the superscript $+$ denotes the positive part. In the output $\mathbf{S}_{(l,c)} \in \mathbb{R}^{(N+1) \times (N+1)}$, each row corresponds to the relevance of the corresponding patch with respect to each of the other patches. Note that (9) reflects the importance 'to-go', i.e., from the current layer until the end of the network. For our method, we only consider the relevance map derived from the row that corresponds to [CLS], that is $\mathbf{S}^{\text{inter}}_{(l,c)} \in \mathbb{R}^{N+1}$. This captures each patch's influence on the classification token for layer $l$ and class $c$. The *inter-layer importance score* $\mathbf{S}^{\text{inter}}_{(l)}$ is obtained by averaging across classes $\mathcal{C}$. Alg. 2 details the aforementioned process.

In Alg. 2, we sample batches for $T$ rounds (step 1) to calculate the inter-layer information ($T = 10$ is used in our experiments). For each image in a batch of data $\mathcal{B}$, we define the set of top-$\gamma$ classes $\mathcal{C}$ ($\gamma = 6$ is used in our experiments). We calculate scores according to steps 9-12: we obtain $\mathbf{R}_{(l,c)}$ in step 9 for all patches (including the [CLS]) for each class $c$ in $\mathcal{C}$ according to the propagation step described in (7), while steps 10-11 compute the scores among all patches (by multiplying $\overline{\mathbf{A}}_{(l,c)}$ from the $l$-th layer to the last layer). Only the first row (corresponding to [CLS]) is kept in step 12

as the importance scores of all patches for each class $c$. The *inter-layer importance score* $\mathbf{S}^{\text{inter}}_{(l)}$ is obtained by averaging across batches and classes $\mathcal{C}$ (step 17), and the number of rounds $T$ (step 21).

## C    DETAILED DERIVATION OF FUSION STEP

This section gives the detailed derivation of equation (4). Recall the problem:

$$\underset{p}{\text{minimize}} \quad \alpha KL\left(p\|p^1\right) + (1-\alpha)KL\left(p\|p^2\right),$$
$$s.t. \quad p_i \geq 0, \ \sum_i p_i = 1,$$

where $\alpha$ ($\in [0,1]$) is a hyperparameter that balances the contribution of intra-/inter-layer scores; $p^1$ is the pmf corresponding to intra-layer scores and $p^2$ is the pmf for inter-layer scores.

First, notice that the constraint $p_i \geq 0$ is implicit (due to the logarithm in the definition of the KL divergence - $KL(p,q) := \sum_i p_i \log \frac{p_i}{q_i}$), so it does not have to be incorporated in the Lagrangian of the problem (i.e., it can be relaxed). The Lagrangian for (3) with multiplier $v$ (corresponding to the equality constraint) is given by:

$$L(p,\lambda,v) = \sum_i \left\{ \alpha p_i \log p_i - \alpha p_i \log p_i^1 + (1-\alpha)p_i \log p_i - (1-\alpha)p_i \log p_i^2 \right\} + v\left(\sum_i p_i - 1\right)$$

$$= \sum_i \left\{ p_i \log p_i - \alpha p_i \log p_i^1 - (1-\alpha)p_i \log p_i^2 + vp_i \right\} - v.$$

The KKT conditions boil down to:

$$\begin{cases} p_i \geq 0, & \forall i \\ \sum_i p_i = 1, & \\ \log p_i + 1 - \log\left(p_i^1\right)^\alpha - \log\left(p_i^2\right)^{1-\alpha} + v = 0. & \forall i \end{cases}$$

It is plain to verify that Slater's condition is satisfied (the explicit constraint set is non-empty and affine) so that strong duality holds. From the last KKT condition, it follows that:

$$\log p_i + 1 - \log\left(p_i^1\right)^\alpha - \log\left(p_i^2\right)^{1-\alpha} + v = 0.$$

Consequently, we can derive:

$$p_i = (p_i^1)^\alpha \cdot (p_i^2)^{(1-\alpha)} \cdot e^{-(1+v)}, \quad \forall i$$

In view of $\sum_i p_i = 1$, we obtain:

$$\sum_i (p_i^1)^\alpha \cdot (p_i^2)^{(1-\alpha)} \cdot e^{-(1+v)} = 1,$$

$$\Rightarrow \quad e^{-(1+v)} = \frac{1}{\sum_i (p_i^1)^\alpha \cdot (p_i^2)^{(1-\alpha)}}. \tag{10}$$

The unique (because of strict convexity) solution is thus given by:

$$p_i = \frac{\left(p_i^1\right)^\alpha \cdot \left(p_i^2\right)^{1-\alpha}}{\sum_i \left(p_i^1\right)^\alpha \cdot \left(p_i^2\right)^{1-\alpha}}, \quad \forall i$$

that is $\mathbf{S}^{\text{fused}} \propto \mathbf{S}^{\text{intra}\,\alpha} \times \mathbf{S}^{\text{inter}\,(1-\alpha)}$.

## D    ADAPTIVE RETENTION RATE SELECTION

We present more details on the design and analysis of *adaptive retention rate selection* (a-STAR, corresponding to **Option 2** in Step 9 of Alg. 1) in this section. We begin by establishing a lemma that is needed to prove that the overall compression of a-STAR is no less than that of STAR, for common parameter $\rho$.

**Lemma 1.** *Consider a pmf that satisfies $p_1 \geq p_2 \geq \cdots \geq p_N$. The CDF $c_i := \sum_{j \leq i} p_i$ satisfies:*

$$c_i \geq \frac{i}{N}, \forall i \in \{1, \ldots, N\}.$$

*Proof.* By definition, $c_N = 1$. Assume $\exists i \in \{1, \ldots, N-1\}$ so that $c_i < \frac{i}{N}$. This implies that $\min_{j \leq i} p_i < \frac{1}{N}$ (since $i \cdot \min_{j \leq i} p_i \leq c_i$). From monotonicity, it holds that $\max_{j > i} p_i \leq \min_{i \leq j} p_i < \frac{1}{N}$. Additionally, $1 = c_N = c_i + \sum_{j > i} p_i < \frac{i}{N} + (N - i)\frac{1}{N} = 1$. Thus, a contradiction. $\square$

The main idea of a-STAR is to use the retention rates of STAR ($p_l$) as weighted percentile values for patch pruning. This leads to a dynamic compression rate (i.e., that depends on the input), since the number of retained patches now depends also on the *distribution* of (fused) scores.

Formally, for the $l$-th Transformer layer ($1 \leq i \leq L$), we obtain a set of normalized fused scores denoted as $\tilde{\mathbf{S}}^{\text{fused}}_{(l)} := \{s^i_l\}^N_{i=1}$. The primary objective is to partition this set into two distinct subsets: $\tilde{\mathbf{S}}^{\text{fused}}_{(l)} = \mathcal{R}_l \sqcup \mathcal{P}_l$, where $\mathcal{R}_l$ represents the retained patches in the $l$-th layer, and $\mathcal{P}_l$ represents the pruned patches in the same layer. The objective is to:

$$\underset{\mathcal{R}_l \subseteq \tilde{\mathbf{S}}^{\text{fused}}_{(l)}}{\text{minimize}} \quad |\mathcal{R}_l| \tag{11}$$

$$\text{s.t.} \quad P(\mathcal{R}_l) \geq p_l,$$

where $P(\mathcal{R}_l) := \sum_{i \in \mathcal{R}_l} s^i_l$. It is plain to check that there exists an optimal solution of (10) where $\mathcal{R}_l$ contains the highest scores (for any solution, switching a score in $\mathcal{R}_l$ that is lower than a score in $\mathcal{P}_l$ would still satisfy the constraint). This suggests that solving (10) can be performed by the steps in Option 2 of Alg. 2 (sorting $\tilde{\mathbf{S}}^{\text{fused}}_{(l)}$ in decreasing order and computing the index where the CDF first exceeds the value $p_l$). Let $p'_l := \frac{1}{N_l}\lceil p_l N_l \rceil \geq p_l$ ($N_l$ denotes the number of patches at layer $l$): note that and solving (10) with $p'_l$ in place of $p_l$ yields an upper bound on the optimal $|\mathcal{R}_l|$. It follows from Lemma 1 that $|\mathcal{R}_l| \leq \lceil p_l N_l \rceil$ (since $c_{\lceil p_l N_l \rceil} \geq p_l$). Since the number of retained patches for STAR is exactly $\lceil p_l N_l \rceil$, we have established that the number of retained patches by a-STAR at layer $l$ is no larger than that for STAR. By repeating this argument layer-wise (in view also of the fact that $N_{l+1} \leq N_l$), we establish that the FLOPs Drop for a-STAR is at least as high as the one for STAR. This is consistent with our experimental results in Table 1. Finally, we can also establish the same monotonicity property for a-STAR as in STAR (i.e., decreasing in ACS and increasing in $\rho$). This is because of the fact that the optimal solution of (10) is non-decreasing in $p_l$.

## E   FULL EXPERIMENTS

In this section, we present the full results of the experiments. Fig. 3 shows the comparison between our approach and baselines on the trade-off between Top-1 Acc. and Throughput on the ImageNet dataset. We measured the throughput of all methods on an NVIDIA A100 GPU with 40GB memory and a batch size of 128. The input image size is $224^2$, and the Multiply-Accumulate computations (MACs) metric is determined using *torchprofile*. Our code was implemented in PyTorch 1.8.0 and Python 3.7 on a system equipped with 4 NVIDIA Tesla A100S GPUs and 8 NVIDIA 3090 GPUs. The CUDA version employed is 11.3, and the inference precision is set to float32. After pruning, the compressed models are fine-tuned for 120 epochs with hard distillation (Touvron et al., 2021) on images of resolution $224^2$ of their corresponding original models. Since ToMe (Bolya et al., 2023) conducts experiments on DeiT model compression by training from scratch for 300 epochs, and Patch Slimming (Tang et al., 2022) does not provide open-source code, we rely directly on the results reported in their respective papers. STAR yields a consistent improvement over the state-of-the-art methods in all scenarios.

**Influence of hyperparameter $\alpha$ in KL fusion:** Table 3 shows the full results of varying the hyperparameter $\alpha$ for KL fusion. In order to experiment with as many choices and models as possible, all experiments listed did not apply fine-tuning. The batch sizes are determined based on the maximum memory allowed by a single A100 GPU for different models. Specifically, the batch sizes are 192, 256, and 512 for DeiT-Base, DeiT-Small, and DeiT-Tiny, while LV-ViT-M and LV-ViT-S employ

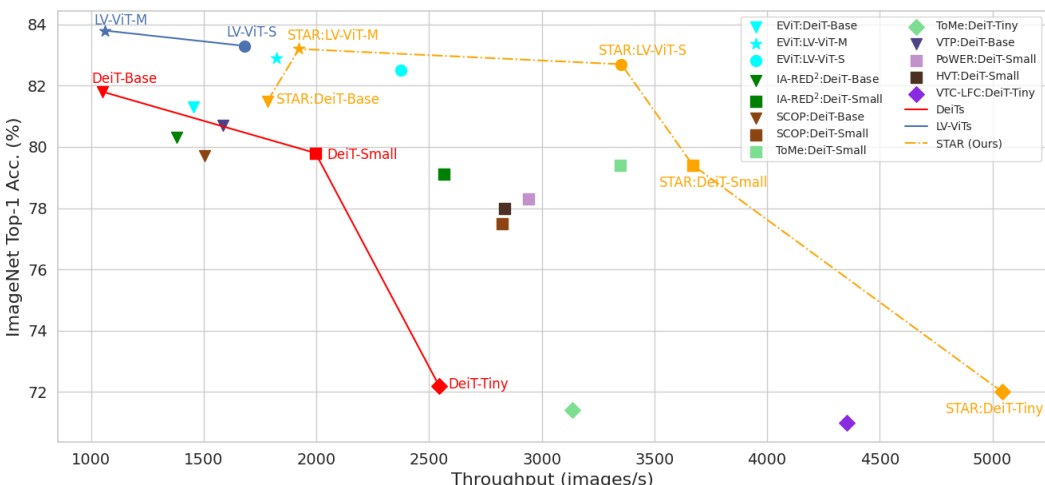

Figure 3: Accuracy-throughput trade-off. The baseline models (DeiT and LV-ViT) are depicted in solid lines, while dashed lines connect points for our pruning method. The same model is denoted by the same marker, while the same method is indicated by the same color. STAR achieves the most favorable trade-off, surpassing other methods in terms of both accuracy and throughput.

Table 3: Effect of hyperparameter $\alpha$ in the KL fusion module of STAR and a-STAR for DeiT and LV-ViT models (without fine-tuning). 'STAR' means using the static retention rate selection while 'a-STAR' refers to using the adaptive retention rate selection. $\alpha = 1$ means intra-layer only and $\alpha = 0$ inter-layer only. The batch size represents the maximum value that can be selected for running the models on a single A100 GPU, and the value of $\rho$ is determined based on the target pruning ratio. The values marked in boldface represent the best results: in all cases, fusion is instrumental (i.e., $\alpha \in \{0, 1\}$ are the worst choices), while there is robustness in choosing $\alpha$ with optimal performance in $[0.15, 0.3]$.

| DeiT-Base | | | KL fusion with Batch size = 192, $\rho = 0.75$ | | | | | | | | | | | |
|---|---|---|---|---|---|---|---|---|---|---|---|---|---|---|
| | $\alpha$ | - | 0.0 | 0.1 | 0.15 | 0.2 | 0.3 | 0.4 | 0.5 | 0.6 | 0.7 | 0.8 | 0.9 | 1.0 |
| STAR | Top-1 Acc. | 81.8 | 77.7 | 79.9 | **80.0** | 79.8 | 79.7 | 79.7 | 79.6 | 79.6 | 79.5 | 79.5 | 79.4 | 79.4 |
| | Top-5 Acc. | 95.6 | 93.2 | 94.5 | **94.7** | 94.6 | 94.4 | 94.5 | 94.4 | 94.4 | 94.3 | 94.3 | 94.2 | 94.1 |
| a-STAR | Top-1 Acc. | 81.8 | 77.4 | 79.4 | **79.7** | 79.6 | 79.5 | 79.3 | 79.3 | 79.2 | 79.2 | 79.2 | 79.2 | 79.1 |
| | Top-5 Acc. | 95.6 | 93.0 | 94.3 | **94.5** | 94.4 | 94.3 | 94.2 | 94.2 | 94.2 | 94.2 | 94.1 | 94.1 | 94.1 |
| DeiT-Small | | | KL fusion with Batch size = 256, $\rho = 0.75$ | | | | | | | | | | | |
| | $\alpha$ | - | 0.0 | 0.1 | 0.15 | 0.2 | 0.3 | 0.4 | 0.5 | 0.6 | 0.7 | 0.8 | 0.9 | 1.0 |
| STAR | Top-1 Acc. | 79.8 | 75.5 | 77.5 | **77.6** | 77.5 | 77.3 | 77.1 | 77.0 | 77.0 | 76.9 | 77.0 | 77.0 | 76.9 |
| | Top-5 Acc. | 95.0 | 92.4 | 93.8 | **93.9** | 93.8 | 93.6 | 93.6 | 93.5 | 93.5 | 93.5 | 93.6 | 93.5 | 93.4 |
| a-STAR | Top-1 Acc. | 79.8 | 75.4 | 77.2 | **77.3** | 77.2 | 77.0 | 76.8 | 76.6 | 76.6 | 76.4 | 76.5 | 76.4 | 76.3 |
| | Top-5 Acc. | 95.0 | 92.4 | 93.6 | **93.7** | 93.6 | 93.4 | 93.4 | 93.3 | 93.3 | 93.3 | 93.2 | 93.2 | 93.1 |
| DeiT-Tiny | | | KL fusion with Batch size = 512, $\rho = 0.8$ | | | | | | | | | | | |
| | $\alpha$ | - | 0.0 | 0.1 | 0.15 | 0.2 | 0.3 | 0.4 | 0.5 | 0.6 | 0.7 | 0.8 | 0.9 | 1.0 |
| STAR | Top-1 Acc. | 72.2 | 66.2 | 68.7 | **68.9** | 68.7 | 68.7 | 68.6 | 68.6 | 68.5 | 68.3 | 68.4 | 68.3 | 68.2 |
| | Top-5 Acc. | 91.1 | 87.5 | 89.0 | **89.1** | 89.0 | 88.9 | 88.9 | 88.9 | 88.8 | 88.8 | 88.8 | 88.7 | 88.8 |
| a-STAR | Top-1 Acc. | 72.2 | 65.7 | 68.4 | **68.6** | 68.3 | 68.3 | 68.2 | 68.2 | 68.0 | 67.9 | 67.9 | 67.8 | 67.6 |
| | Top-5 Acc. | 91.1 | 87.2 | 88.6 | **88.8** | 88.7 | 88.6 | 88.6 | 88.5 | 88.5 | 88.5 | 88.4 | 88.4 | 88.3 |
| LV-ViT-M | | | KL fusion with Batch size = 128, $\rho = 0.8$ | | | | | | | | | | | |
| | $\alpha$ | - | 0.0 | 0.1 | 0.2 | 0.25 | 0.3 | 0.4 | 0.5 | 0.6 | 0.7 | 0.8 | 0.9 | 1.0 |
| STAR | Top-1 Acc. | 83.8 | 80.3 | 82.1 | 82.2 | 82.3 | **82.4** | 82.3 | 82.2 | 82.2 | 82.1 | 82.0 | 82.0 | 81.9 |
| | Top-5 Acc. | 96.6 | 94.7 | 95.8 | 95.9 | 95.9 | **96.0** | 95.9 | 95.9 | 95.8 | 95.8 | 95.8 | 95.8 | 95.7 |
| a-STAR | Top-1 Acc. | 83.8 | 79.8 | 81.9 | 81.9 | 82.0 | **82.1** | 81.9 | 81.7 | 81.8 | 81.7 | 81.6 | 81.6 | 81.6 |
| | Top-5 Acc. | 96.6 | 94.3 | 95.7 | 95.7 | 95.7 | **95.8** | 95.7 | 95.6 | 95.6 | 95.5 | 95.5 | 95.5 | 95.4 |
| LV-ViT-S | | | KL fusion with Batch size = 256, $\rho = 0.65$ | | | | | | | | | | | |
| | $\alpha$ | - | 0.0 | 0.1 | 0.2 | 0.25 | 0.3 | 0.4 | 0.5 | 0.6 | 0.7 | 0.8 | 0.9 | 1.0 |
| STAR | Top-1 Acc. | 83.3 | 71.5 | 78.4 | 78.6 | 78.6 | **78.7** | 78.5 | 78.4 | 78.1 | 78.2 | 78.0 | 77.9 | 78.0 |
| | Top-5 Acc. | 96.3 | 89.6 | 93.7 | 93.7 | 93.7 | **93.8** | 93.7 | 93.5 | 93.5 | 93.4 | 93.3 | 93.3 | 93.3 |
| a-STAR | Top-1 Acc. | 83.3 | 71.2 | 77.9 | 78.0 | **78.1** | 78.0 | 77.9 | 77.7 | 77.8 | 77.6 | 77.6 | 77.5 | 77.3 |
| | Top-5 Acc. | 96.3 | 89.5 | 93.3 | **93.4** | **93.4** | 93.3 | 93.2 | 93.1 | 92.9 | 92.9 | 92.8 | 92.7 | 92.7 |

batch sizes of 128 and 256, respectively. For DeiT-Base, DeiT-Small, and DeiT-Tiny models, the highest Top-1 Acc. and Top-5 Acc. are achieved with $\alpha$ set to 0.15, while for the LV-ViT-M and LV-ViT-S model, the optimal values are 0.3 and 0.25, respectively. It is noteworthy that after applying STAR to prune DeiT-Small, the resulting Top-1 Acc. and Top-5 Acc. are improved by 0.7% and 0.5%, respectively, compared to using only intra-layer importance scores. This finding further supports the effectiveness of our fusion module.

**Influence of hyperparameter $\rho$ on pruning:** Fig. 4(a) illustrates the impact of $\rho$ on the layer-wise retention rate of DeiT-Base generated by the static retention rate selection method described in Sec. 5.2. It illustrates the previously discussed monotonicity property (i.e., the higher the $\rho$, the higher the retention rate at each layer). Table 4 and Table 5 show the accuracy of DeiT and LV-ViT models varying with GFLOPs under different compression ratios without fine-tuning; Figure 4(b) visualizes it for the DeiT-Base model. For instance, in DeiT-Base, when the value of $\rho$ is set to 0.9, the corresponding GFLOPs of the pruned DeiT-Base model is 14.0. As $\rho$ decreases to 0.6, the GFLOPs also decrease to 7.7. This demonstrates the effectiveness of our approach in that a single hyperparameter $\rho$ can control the overall compression. In Figure 4(c), we compare the retention rate calculated by STAR (using the *static retention rate selection*) and a-STAR (using the *adaptive retention rate selection*). As expected, the CDF curve is concave, which shows that the retention rate calculated by a-STAR is no larger than that for STAR ($p_l$), and this is consistent with our discussion in Appendix D.

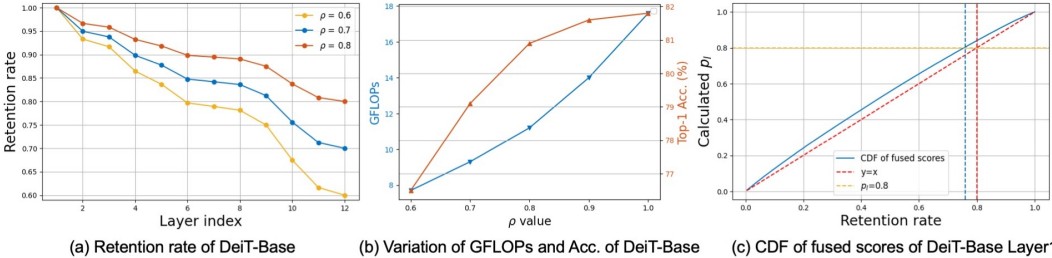

(a) Retention rate of DeiT-Base    (b) Variation of GFLOPs and Acc. of DeiT-Base    (c) CDF of fused scores of DeiT-Base Layer1

Figure 4: (a) Impact of $\rho$ on the layer-wise static retention rate selection. (b) Impact of $\rho$ on the GFLOPs and Top-1 Acc. of Deit-Based using STAR. (c) Comparison of the retention rate derived by STAR and a-STAR.

Table 4: DeiT results under different compression levels (without fine-tuning).

| Model | $\rho$ | GFLOPs | Top-1 Acc. | Top-5 Acc. |
|---|---|---|---|---|
| | 1.0 | 17.6 | 81.8 | 95.6 |
| | 0.9 | 14.0 | 81.6 | 95.5 |
| DeiT-Base | 0.8 | 11.2 | 80.9 | 95.1 |
| | 0.7 | 9.3 | 79.1 | 94.3 |
| | 0.6 | 7.7 | 76.5 | 92.8 |
| | 1.0 | 4.6 | 79.8 | 95.0 |
| | 0.9 | 3.6 | 79.4 | 94.8 |
| DeiT-Samll | 0.8 | 2.8 | 78.7 | 94.2 |
| | 0.7 | 2.3 | 76.4 | 93.1 |
| | 0.6 | 2.0 | 73.5 | 92.6 |
| | 1.0 | 1.3 | 72.2 | 91.1 |
| | 0.9 | 0.9 | 71.3 | 90.7 |
| DeiT-Tiny | 0.8 | 0.7 | 68.9 | 89.1 |
| | 0.7 | 0.6 | 65.8 | 87.5 |
| | 0.6 | 0.5 | 63.4 | 84.9 |

Table 5: LV-ViT results under different compression levels (without fine-tuning).

| Model | $\rho$ | GFLOPs | Top-1 Acc. | Top-5 Acc. |
|---|---|---|---|---|
| | 1.0 | 12.6 | 83.8 | 96.6 |
| | 0.9 | 8.8 | 83.6 | 96.6 |
| LV-ViT-M | 0.8 | 6.7 | 82.4 | 96.0 |
| | 0.7 | 5.5 | 80.3 | 95.2 |
| | 0.6 | 4.7 | 77.3 | 91.5 |
| | 1.0 | 6.5 | 83.3 | 96.3 |
| | 0.9 | 5.0 | 83.1 | 96.3 |
| LV-ViT-S | 0.8 | 4.0 | 82.1 | 95.9 |
| | 0.7 | 3.4 | 80.3 | 94.8 |
| | 0.6 | 3.0 | 77.5 | 92.9 |

**Influence of data size used to calculate ACS on pruning:** In this ablation study, we use different sizes of data to calculate the average cosine similarity between patches on DeiT-Base using STAR and show its impact on model accuracy with the use of $\rho$=0.75. The result in Table 6 shows that when the data size is 500, the pre-computing takes 14.2 seconds and the Top-1 Acc. and Top-5 Acc. are 0.1% and 0.2% better than using data size 250, while it keeps the same when increasing the data size to 1000. This result supports the choice of using a data size of 500 in our experiment.

**Impact of fine-tuning:** The result in Table 7 displays the impact of fine-tuning on DeiT and LV-ViT after-pruned models accuracy for STAR. It shows that fine-tuning can enhance model accuracy by a minimum of 0.4% and a maximum of 4%.

Table 6: Impact of data sizes used to calculate the average cosine similarity between patches on the DeiT-Based model accuracy when using $\rho$=0.75 without fine-tuning for STAR. Data Size = 500 gives the best trade-off between pre-computing time and model accuracy.

| Data Size | Time of Computing ACS | GFLOPs | Top-1 Acc. | Top-5 Acc. |
|---|---|---|---|---|
| 1000 | 27.6s | 10.22 | 80.0 | 94.7 |
| 500 | 14.2s | 10.20 | 80.0 | 94.7 |
| 250 | 7.3s | 10.20 | 79.9 | 94.5 |
| 100 | 3.4s | 10.18 | 79.8 | 94.5 |

Table 7: Impact of fine-tuning on accuracy for STAR. FT denotes fine-tuning the compressed model for 120 epochs. w/o means without fine-tuning, w/ means with fine-tuning, and $\uparrow$ means the improvement of Acc. under fine-tuning.

| Model | GFLOPs | Top-1 Acc. / Top-5 Acc. | | |
|---|---|---|---|---|
| | | w/o FT | w/ FT | $\uparrow$ |
| DeiT-Base | 10.2 | 80.0/94.7 | 81.5/95.5 | 1.5/0.8 |
| DeiT-Samll | 2.6 | 77.6/93.9 | 79.4/94.5 | 1.8/0.6 |
| DeiT-Tiny | 0.7 | 68.9/89.1 | 72.0/90.7 | 3.1/1.6 |
| LV-ViT-M | 6.7 | 82.4/96.0 | 83.3/96.4 | 0.9/0.4 |
| LV-ViT-S | 3.2 | 78.7/93.8 | 82.7/96.2 | 4.0/2.4 |

Table 8: Comparison between STAR and DiffRate with a fine-tuning of 30 epochs.

| DeiT-Base | GFLOPs | Acc. (%) | Throughput | Speedup (%) |
|---|---|---|---|---|
| Baseline | 17.6 | 81.8 | 1,051.6 | - |
| DiffRate | 11.5 | 81.6 | 1,444.9 | 37.4 |
| STAR | 11.2 | 81.6 | 1,595.9 | 51.7 |
| a-STAR | 11.0 | 81.5 | 1,702.1 | 61.8 |

Table 9: DeiT performance on out-of-distribution datasets (ImageNet-R).

| Model | Top-1 Acc. on the before-pruned model (%) | GFLOPs | Top-1 Acc. on the after-pruned model (%) | GFLOPs | FLOPs $\downarrow$ (%) |
|---|---|---|---|---|---|
| DeiT-Base | 45.36 | 17.6 | 43.64 | 11.2 | 36.4 |
| DeiT-Samll | 42.50 | 4.6 | 41.05 | 2.8 | 39.1 |
| DeiT-Tiny | 33.19 | 1.3 | 31.07 | 0.7 | 46.1 |

**Comparison with DiffRate:** Result in Table 8 is about the comparison between STAR and DiffRate (Chen et al., 2023) on DeiT-Base with a fine-tuning of 30 epochs under the same setting as DiffRate. The throughput of this experiment was tested on an NVIDIA A100 GPU with 40GB memory, utilizing the batch size 128 and fp32. With the same Top-1 Acc. of the after-pruned DeiT-Base, STAR achieves a reduction of 0.3 in GFLOPs and +14.3% throughput speedup compared to DiffRate, while a-STAR achieves 24.4% higher speedup with negligible accuracy drop (-0.1%)

**Model performance on out-of-distribution datasets:** The result in Table 9 lists the Top-1 Acc. of the before-pruned model and after-pruned model (without fine-tuning) on the out-of-distribution dataset ImageNet-R (Hendrycks et al., 2021). It is worth noting that for DeiT-Small, STAR achieves a reduction of 39.1% in FLOPs, with a drop of only 1.45% in Top-1 Acc. on ImageNet-R. These results demonstrate the robustness of STAR as an effective patch pruning method on the out-of-distribution dataset.

