# OpenReview forum: "Synergistic Patch Pruning for Vision Transformer: Unifying Intra- & Inter-Layer Patch Importance"
_ICLR.cc/2024/Conference — ICLR 2024 poster_

### Official Review · Reviewer_fwdP · 2023-10-24

**Soundness:** 3 good
**Presentation:** 3 good
**Contribution:** 3 good
**Rating:** 6
**Confidence:** 4

**Summary:**

This paper proposes a dynamic patch pruning method for accelerating vision transformers. Inter-layer importance scores are computed to obtain the importance of patches at a specific layer. The inter-layer importance scores consider the impact of patches across layers on the final classification output.

**Strengths:**

The paper articulates its points effectively.
The target issues of the paper are meaningful and worth exploring. The idea is novel, and the motivation is clear.
The inclusion of experimental analysis on pruning vision transformers is a strength of the paper. This analysis provides valuable insights and promotes understanding of the proposed method.

**Weaknesses:**

1. This paper does not cite some dynamic pruning papers, such as [1,2,3].
[1] Manifold Regularized Dynamic Network Pruning. CVPR2021
[2] DYNAMIC MODEL PRUNING WITH FEEDBACK. ICLR2020
[3] Dynamic Network Pruning with Interpretable Layerwise Channel Selection. AAAI2020

2. The patch pruning methods cannot save the parameters, while channel pruning can achieve this goal. How to reduce the parameters is worth exploring.

**Questions:**

See Weaknesses

---

> ### Author Response · Authors · 2023-11-21
> **Response to Reviewer fwdP**
>
> Thank you very much for taking the time to review our paper and provide very useful comments. We have revised the paper to incorporate all of your suggestions.
>
> > W1: This paper does not cite some dynamic pruning papers, such as [1,2,3]. [1] Manifold Regularized Dynamic Network Pruning. CVPR2021 [2] DYNAMIC MODEL PRUNING WITH FEEDBACK. ICLR2020 [3] Dynamic Network Pruning with Interpretable Layerwise Channel Selection. AAAI2020
>
> Thank you for providing these references which we list in the revised paper for a more complete discussion of the literature: please refer to page 1, Introduction, paragraph 1, line 9.
>
> > W2: The patch pruning methods cannot save the parameters, while channel pruning can achieve this goal. How to reduce the parameters is worth exploring.
>
> Thank you for this comment. You are right that patch pruning does not operate by pre-reducing network parameters (in a static fashion) like channel pruning which is the customary approach, for example in CNNs. In ViTs, the process of pruning entire channels can result in information loss due to its unique structure, where input images are divided into multiple patches, and computations are carried out in parallel across all patches. This represents a semantic mechanism tailored specifically for ViTs. The significance of information associated with each image patch varies, and not all patches are equally crucial for task fidelity. Consequently, pruning patches becomes a means to save computational resources. Importantly, patch pruning is adaptive, as its effectiveness relies on the input content, and it gives greater flexibility by allowing the dynamic selection of a subset of patches deemed useful for each layer. Our paper focuses solely on patch pruning, and we have added this information in the second paragraph of Section 2 on page 3 to make it clear.
>
> Additionally, we have conducted comparison experiments with the state-of-the-art channel pruning methods named SCOP and VTP, specifically tailored for ViTs. The results presented in Table 1 demonstrate the superior performance of STAR. Notably, STAR achieved a 1.9% improvement in Top-1 Accuracy under the same model size and a throughput acceleration of up to 1.3x for the DeiT-Small model compared to SCOP. In comparison to VTP applied to DeiT-Base, STAR exhibited a remarkable +34.1% speedup and a 0.3% improvement in Top-1 Accuracy, while maintaining a lower computational cost of 9.3 GFLOPs.
>
> Thank you once again for your valuable suggestions and comments.

---

### Official Review · Reviewer_cb4H · 2023-10-30

**Soundness:** 2 fair
**Presentation:** 2 fair
**Contribution:** 2 fair
**Rating:** 6
**Confidence:** 4

**Summary:**

This paper suggests a token compression method used for ViTs named STAR. To better determine which tokens should be pruned, STAR combines an intra-layer importance metric given by [CLS] token and an inter-layer importance metric given by layer-wise relevance propagation into a more comprehensive metric. STAR is able to automate pruning ratio selection across layers. Experimental results on DeiT and LV-ViT show improved performance over some previous token compression methods.

**Strengths:**

This paper is generally well-written.  Experimental results on DeiT and LV-ViT show improved performance over some previous token compression methods.

**Weaknesses:**

- Novelty
  - The proposed STAR is a combination of existing importance-aware (given by [CLS] token) token compression methods such as EViT[1] and layer-wise relevance propagation method[2]. Simply composing two existing metrics, as the core novelty of the paper, does not deserve a good contribution, in my opinion.
  - The proposed STAR is not the first work to capture the cross-layer importance and enables different compression ratios among different layers. For example, DiffRate[3] can also automatically learn different compression rates across layers. The lack of thorough comparisons with these previous approaches also harms the contribution and novelty of this paper.
- Method
  - The proposed STAR introduces an extra procedure for pre-computing average cosine similarity as a metric, while other token compression method does not. It is unclear how much extra time this procedure will cost and how the size of the dataset used for pre-computing will affect the model performance.
- Experiments
  - Some important baselines are missing. For example, DiffRate[3] mentioned above.
  - The setting of comparison experiments is unfair. STAR uses an extra distillation to enhance the after-pruning performance, while some compared methods, such as ToMe[4], do not use it. Experimental results without distillation should also be reported to achieve a fair comparison.
  - Some ablation studies are missing. For example, the impact of the dataset size used for pre-computing metrics on model performance. Besides, it would be better if there were zero-shot experiments to investigate how well these pre-computed metrics perform on out-of-distribution datasets (e.g., ImageNet-R[5]).
  - Experimental results at different compression ratios on the same model are missing. For each model, there is only one compression ratio used, and different compression ratios on the same model are needed to verify the effectiveness of the proposed STAR under higher and lower compression ratios.
  - Experiments on different architectures are insufficient. The experiments are only conducted on DeiT and LV-ViT, and it would be better to also have experimental results on other branches of the ViT family, for example, Swin Transformer[6] and CSWin Transformer[7], which adopt window attentions.

References
[1] EViT: Expediting Vision Transformers Via Token Reorganizations.
[2] Transformer interpretability beyond attention visualization.
[3] DiffRate: Differentiable Compression Rate for Efficient Vision Transformers.
[4] Token merging: Your vit but faster.
[5] The many faces of robustness: A critical analysis of out-of-distribution generalization.
[6] Swin transformer: Hierarchical vision transformer using shifted windows.
[7] CSWin Transformer: A General Vision Transformer Backbone With Cross-Shaped Windows.

**Questions:**

See Weaknesses.

---

> ### Author Response · Authors · 2023-11-21
> **Response to Reviewer cb4H (part 1/2)**
>
> We would like to thank you for your constructive comments, which have been especially helpful in improving the paper! We have addressed all your comments carefully, and below we include point-to-point responses to each question/comment that you raised. We have also modified the paper accordingly (changes are marked in blue).
>
> **Response to Weaknesses - Novelty1:**
>
> Thank you for your comments. We would like to first clarify where the novelty of our paper lies.
>
> You are absolutely right that these techniques are available in the literature. The novelty of our work is not about combining methods, rather it is about asking the question of inter-layer importances. While LRP is a tool used for model interpretability, it has not been invoked for this problem. The other two key novelty points are the fusion mechanism for inter-layer and intra-layer importances, and a new auto-tuning mechanism for the layer-wise compression rates with no overhead of re-training.
>
> We have added a summary of the main novelty points on page 2 before the contributions.
>
> **Response to Weaknesses - Novelty2:**
>
> Thanks for bringing this very interesting paper to our attention. We have included it in our literature review and examined the differences as follows:
>
> DiffRate automatically selects compression rates by incorporating a sparsity-promoting regularizer into the loss function controlled by a hyperparameter $\lambda$. Nevertheless, it requires retraining for different compression targets and does not address inter-layer dependencies explicitly. In STAR, the auto-tuning mechanism for layer-wise compression rates incurs no overhead in terms of retraining: this uses offline metrics (ACS) and contains a single hyperparameter ($\rho$) to control the overall compression. Adjusting $\rho$ and performing online pruning (at inference) yields results quickly.
>
> We have added this comparison to the third paragraph of Section 2, page 3 to highlight the differences.
>
> **Response to Weaknesses - Method:**
>
> Thank you for your questions. In our experiment, we used 500 images to calculate the average cosine similarity between patches, which takes about 14.2 seconds of pre-computing. Regarding the influence of dataset size used for pre-computing on model accuracy, we have added a new ablation study in Table 6, on Appendix E, page 19 to show its impact on the DeiT-Based model accuracy when using $\rho$=0.75 on STAR without fine-tuning. The result shows that when data size = 500, it gives the best trade-off between pre-computing time and model accuracy.
>
> | Data Size | Time of Computing ACS | GFLOPs | Top-1 Acc. | Top-5 Acc. |
> | :-------: | :-------------------: | :----: | :--------: | :--------: |
> |   1000    |         27.6s         | 10.22  |    80.0    |    94.7    |
> |    500    |         14.2s         | 10.20  |    80.0    |    94.7    |
> |    250    |         7.3s          | 10.20  |    79.9    |    94.5    |
> |    100    |         3.4s          | 10.18  |    79.8    |    94.5    |
>
> We also added one sentence to clarify the data size that we used for the pre-computing metric in the second paragraph of Section 6.2, page 9.
>
> **Response to Weaknesses - Experiments1:**
>
> We appreciate the reviewer's understanding that it is not plausible to re-test all scenarios for DiffRate. We have added a new experiment in Table 8, on Appendix E, page 19 about the comparison between STAR and DiffRate [2] on DeiT-Base with a fine-tuning of 30 epochs under the same setting as DiffRate. The throughput of this experiment was tested on an NVIDIA A100 GPU with 40GB memory, utilizing the batch size 128 and fp32 for each model. With the same Acc. of the after-pruned DeiT-Base, STAR achieves a reduction of 0.3 in GFLOPs and +14.3% throughput speedup compared to DiffRate, while a-STAR achieves 24.4% higher speedup with negligible accuracy drop (-0.1%)
>
> | DeiT-Base | GFLOPs | Acc. (%) | Throughput | Speedup (%) |
> | :-------: | :----: | :------: | :--------: | :---------: |
> | Baseline  |  17.6  |   81.8   |  1,051.6   |      -      |
> | DiffRate  |  11.5  |   81.6   |  1,444.9   |    37.4     |
> |   STAR    |  11.2  |   81.6   |  1,595.9   |    51.7     |
> |  a-STAR   |  11.0  |   81.5   |  1,702.1   |    61.8     |

---

> ### Author Response · Authors · 2023-11-21
> **Response to Reviewer cb4H (part 2/2)**
>
> **Response to Weaknesses - Experiments2:**
>
> Thank you very much for raising this concern. The fine-tuning setting with hard distillation is consistent across our experiments for all post-pruned methods, with the exception of ToMe (Bolya et al., 2023) and Patch Slimming (PS) (Tang et al., 2022). Our approach aligns with the settings employed in prior research studies (Wang et al., 2022; Zheng et al., 2022; Touvron et al., 2021) and we directly use the pre-trained models. Given that ToMe conducts experiments on DeiT model compression by training from scratch for 300 epochs, and Patch Slimming does not provide open-source code, we rely directly on the results reported in their respective papers.
>
> To enhance transparency, we have incorporated this information in the first paragraph Appendix E on page 16 to provide clear clarification.
>
> **Response to Weaknesses - Experiments3:**
>
> We have added a new ablation study in Table 6, on Appendix E, page 19 to show the influence of dataset size used for pre-computing on model performance without fine-tuning (as described in the response to Weakness - Method). The result shows that when data size = 500, it gives the best trade-off between pre-computing time and model accuracy.
>
> Thank you for your suggestion! We added a new experiment to test the performance of the after-pruned DeiT model on the out-of-distribution dataset ImageNet-R [1] to show the effectiveness of the pre-computed metrics (Table 9 on Appendix E, page 19). We test the accuracy of the before-pruned model and after-pruned DeiT-Base model (without fine-tuning) directly [1]. It is worth noting that for DeiT-Small, STAR achieves a  reduction of 39.1% in FLOPs, with a drop of only 1.45% in Top-1 Acc. on ImageNet-R. These results demonstrate the robustness of STAR as an effective patch pruning method on the out-of-distribution dataset.
>
> |   Model    | Top-1 Acc. on the before-pruned model (%) | GFLOPs | Top-1 Acc. on the after-pruned model (%) | GFLOPs | FLOPs $\downarrow$ (%) |
> | :--------: | :---------------------------------------: | :----: | :--------------------------------------: | :----: | :--------------------: |
> | DeiT-Base  |                   45.36                   |  17.6  |                  43.64                   |  11.2  |          36.4          |
> | DeiT-Small |                   42.50                   |  4.6   |                  41.05                   |  2.8   |          39.1          |
> | DeiT-Tiny  |                   33.19                   |  1.3   |                  31.07                   |  0.7   |          46.1          |
>
> **Response to Weaknesses - Experiments4:**
>
> Thank you very much for your question. We have incorporated new experiments to assess the model performance under various compression ratios. Here, we provide detailed results for DeiT-Base due to the limit of characters. The complete set of results is included in Tables 4&5 and Figure 4(b) (Appendix E on page 18). The results of this experiment were obtained without fine-tuning in view of the stringent time constraints of the rebuttal period.
>
> Table 4 and Table 5 show the accuracy of DeiT and LV-ViT models varying with GFLOPs under different compression ratios without fine-tuning; Figure 4(b) visualizes it for the DeiT-Base model. For instance, when  $\rho$ is 0.9, the corresponding GFLOPs of the after-pruned DeiT-Base model is 14.0. As $\rho$ decreases to 0.6, the GFLOPs also decrease to 7.7. This demonstrates the effectiveness of our approach in that a single hyperparameter $\rho$ can control the overall compression.
>
> | $\rho$ | GFLOPs | Top-1 Acc. | Top-5 Acc. |
> | :----: | :----: | :--------: | :--------: |
> |  1.0   |  17.6  |    81.8    |    95.6    |
> |  0.9   |  14.0  |    81.6    |    95.5    |
> |  0.8   |  11.2  |    80.9    |    95.1    |
> |  0.7   |  9.3   |    79.1    |    94.3    |
> |  0.6   |  7.7   |    76.5    |    92.8    |
>
> **Response to Weaknesses - Experiments5:**
>
> We appreciate your suggestions.  We would like to point out that patch pruning is not suitable for window-based attention models like the Swin Transformer. This is due to their utilization of a hierarchical architecture that relies on patch downsampling [3]. Pruning patches based on their importance in each layer would compromise the spatial integrity of the feature map essential to the hierarchical architecture [2].
>
> Again, thank you for your suggestions and comments.
>
> **References:**
>
> [1] The many faces of robustness: A critical analysis of out-of-distribution generalization.
>
> [2] DiffRate: Differentiable Compression Rate for Efficient Vision Transformers.
>
> [3] VTC-LFC: Vision Transformer Compression with Low-Frequency Components.

---

> > ### Comment · Reviewer_cb4H · 2023-11-23
> >
> > Thanks for the detailed response, and I have carefully read all responses. I have raised my score since most of my concerns are properly addressed.

---

### Official Review · Reviewer_kdgo · 2023-11-01

**Soundness:** 3 good
**Presentation:** 3 good
**Contribution:** 3 good
**Rating:** 8
**Confidence:** 4

**Summary:**

This paper presents a new token pruning method for vision transformers. It utilizes an online evaluation of intra-layer importance and an offline evaluation of inter-layer importance of each token (patch), using a newly designed method named Layer-wise Relevance propagation. The patches are pruned at each layer by maintaining only the top-k important ones. It also uses the average cosine similarity to decide the pruning ratio of each layer. It also introduces another dynamic pruning ratio selection method based on the input features. The method is evaluated on ImageNet classification with DeiT and LV-ViT. It outperforms existing token pruning methods on Transformers such as SCOP, PoWER, ToMe, VTC-LFC etc with better accuracy under similar compression ratio.

**Strengths:**

-	The idea of using attention score with the CLS token as the cues for token pruning is simple and effective. Combining the intra-layer importance and inter-layer importance is novel. The proposed method of using average cosine similarity to decide the pruning ratio facilitates automatic estimation of pruning ratio at each layer.
-	The conducted experiments cover different baseline methods and different pruning ratios, showing the proposed method can outperforming the existing patch-pruning methods.
-	Paper writing is clear and easy to follow.

**Weaknesses:**

-	It seems the pruned model still needs finetuning with 120 epochs to reach a strong performance. Is this setting similar to the existing methods? If the finetuning settings of different methods are different, it will lead to an unfair comparison.
-	In the fusion of intra and inter-layer importance scores, a parameter $\alpha$ is used. How is this parameter tuned? Do you use the same \alpha for different models?
-	Some parts of the paper are too verbose. Abstract includes too much technical details. The claimed contribution in the introduction section is too long and is hard to understand (including too many details).

**Questions:**

-	How is the throughput measured (Table 1). What kind of engineering setup do you use? Such as usage of TensorRT, cuda version, inference precision (fp16 or float32). What is the batch size used (batch size has a large impact on throughput).

---

> ### Author Response · Authors · 2023-11-21
> **Response to Reviewer kdgo**
>
> Thank you very much for your positive reception and constructive comments, which we address in the following point-by-point.
>
> > W1: It seems the pruned model still needs finetuning with 120 epochs to reach a strong performance. Is this setting similar to the existing methods? If the finetuning settings of different methods are different, it will lead to an unfair comparison.
>
> Yes, you are right, the setting is common across baselines.  Thank you for raising this question, and we fully agree that the finetuning setting should be explicated in the paper. For this purpose, we have added the following sentence in the first paragraph of Appendix E on page 16:
>
> - After pruning, the compressed models are fine-tuned for 120 epochs with hard distillation (Touvron et al., 2021) on images of resolution $224^2$ of their corresponding original models.
>
> > W2: In the fusion of intra and inter-layer importance scores, a parameter $\alpha$  is used. How is this parameter tuned? Do you use the same $\alpha$   for different models?
>
> Thank you for your question. Due to length constraints, this ablation study is presented in Appendix E, Table 3 on page 17. A search method was used to adjust the value of $\alpha$. As can be seen from Table 3, a selection between 0.15 and 0.3 of the optimal $\alpha$ value is robust. For each experiment in the paper, we use the best value from Table 3. To enhance clarity, we have added the following sentence to Section 6.2, paragraph 4 on page 9 in blue:
>
> - We use $\alpha = 0.15$ in DeiT models and $0.3$  for LV-ViT models. Details of the selection of the hyperparameter $\alpha$ in the KL fusion module of STAR are listed in Table 3 in Appendix E.
>
> > W3: Some parts of the paper are too verbose. Abstract includes too much technical details. The claimed contribution in the introduction section is too long and is hard to understand (including too many details).
>
> Thank you for your comment. We agree with you, and we have made the following changes in the paper (marked in blue in the revised paper) to enhance the clarity and readability:
>
> - We have trimmed the abstract to omit too technical details about the "how".
> - In contribution, we have simplified the description of our contributions and summarized the novelties on page 2.
>
> > Q: How is the throughput measured (Table 1). What kind of engineering setup do you use? Such as usage of TensorRT, cuda version, inference precision (fp16 or float32). What is the batch size used (batch size has a large impact on throughput).
>
> Thank you for your question. We have included a sentence in the first paragraph of Section 6, line 8 on page 7, and state the details in Appendix E on page 16 to make this explicit.
>
> - We measured the throughput of all methods on an NVIDIA A100 GPU with 40GB memory and a batch size of 128. The input image size is $224^2$, and the Multiply-Accumulate computations (MACs) metric is determined using *torchprofile*. Our code was implemented in PyTorch 1.8.0 and Python 3.7 on a system equipped with 4 NVIDIA Tesla A100S GPUs and 8 NVIDIA 3090 GPUs. The CUDA version employed is 11.3, and the inference precision is set to float32.
>
> Once again, thank you very much for your suggestions and comments that have helped to enhance the readability and clarity of the paper.

---

> > ### Comment · Reviewer_kdgo · 2023-11-22
> >
> > The authors have addressed most of my concerns. I will keep my original rating.

---

### Official Review · Reviewer_h5Sc · 2023-11-01

**Soundness:** 3 good
**Presentation:** 1 poor
**Contribution:** 4 excellent
**Rating:** 6
**Confidence:** 4

**Summary:**

The paper proposes a new patch pruning method (STAR) by combining intra-layer and inter-layer scores.
The intra-layer score uses attention values to CLS token, and the inter-layer score utilizes Layer-wise relevance Propagation (LRP) which was proposed in a previous study for ViT interpretability. For inference, the intra-layer score is calculated by the network in an online manner, while a saved inter-layer score which is averaged over training data is used to utilize it without additional computation. By well-designed pruning threshold determination method, STAR outperforms other patch pruning methods.

**Strengths:**

- STAR shows that offline importance score—saved inter-layer score statistics over training data—can be effectively used for patch pruning methods. I believe this is the paper's key contribution.

- STAR achieves reasonable performance improvement over other baselines.

- STAR enables adaptive inference, which gives additional performance gain.

- A hyper-parameter for compression rate handling is well-designed

**Weaknesses:**

- Writing is unclear and hard to understand.

- Method is incremental. STAR is a combination of two methods: attention score for CLS token and Layer-wise Relevance Propagation (LRP) for ViT. In particular, LRP for STAR looks the same as the original LRP for ViT.

**Questions:**

- Is there any difference from LRP for STAR compared to [A]?
  - [A] Transformer Interpretability Beyond Attention Visualization, CVPR 2021

- As I understand, LRP in [A] requires a relevance score back-propagation for every computation unit. But, Line 9 in Algorithm 1 is not enough to explain it. I strongly recommend authors redesign Section 4 to help readers to understand LRP. Current Algorithm 1 has a lot of flaws and errors. If it is not different from [A], I recommend simplifying the explanation and focusing on explaining the inference stage.

- I think LRP statistics on the training set are used for inference, like batch-norm. Am I right?
It is an important point but not clearly described.
Please clarify the inference process. Especially, it should be clearly stated to check whether additional computation for LRP is excluded from inference costs or not.
If LRP computation is required for every inference, I will lower my rating.

- What is the impact of the fine-tuning process? Can STAR work without fine-tuning?

- Could you visualize saved training stats of inter-layer scores (LRP)? Is there any pattern related to CLS token?

---

> ### Author Response · Authors · 2023-11-21
> **Response to Reviewer h5Sc (part 1/2)**
>
> Thank you very much for your time and great efforts in reviewing our paper. We have revised the paper to address all of your comments and incorporate all of your suggestions.
>
> >  W1: Writing is unclear and hard to understand.
>
> Thank you very much for your comment. We acknowledge that some parts were verbose and technical, therefore, we have simplified the abstract on page 1, the contributions on page 2, and added a description of the key novelty points on page 2, where we have de-emphasized technical details.
>
> In addition, we have relocated Algorithm 1 in Section 4 and its description to Appendix B on page 13. In the revised Section 4, we include a motivating figure for the need for inter-layer analysis as well as an overview of the slight modification of the LRP method (last paragraph of page 4).
>
> > W2: Method is incremental. STAR is a combination of two methods: attention score for CLS token and Layer-wise Relevance Propagation (LRP) for ViT. In particular, LRP for STAR looks the same as the original LRP for ViT.
>
> Thank you for your important comment. In the revised paper, we have added a sentence before the contributions on page 2 to clearly explain the novelty of this paper, which is not just the combination of two existing methods. In particular, there are three main novel aspects: a) asking the question of how **inter-layer importances** of patches can serve to improve the compression (to the best of our knowledge, this has not been addressed before and the (modified) LRP is just the tool to accomplish this - notice that LRP was introduced for interpretability, not for the purpose used in this paper), b) **Fusion** mechanism for inter-layer and intra-layer importances, and c) a new **auto-tuning** mechanism for the layer-wise compression rates with no overhead of re-training.
>
> > Q1: Is there any difference from LRP for STAR compared to [A]?
> >
> > ​        [A] Transformer Interpretability Beyond Attention Visualization, CVPR 2021
>
> Yes, there are some differences, but these are not major and the method (LRP) is not a novelty point (the novelty lies in the question of inter-layer importances rather than the adopted tool). For this reason, we have explicated this in the paper (Section 4 on page 4) and have moved Algorithm 1 and its description to Appendix B on page 13. In the revised Section 4, we also explain the modifications in our proposed solution:
>
> - Instead of operating at the pixel level, we carry an additional aggregation of multiple pixels' relevance belonging to the same patch.
> - Rather than focusing solely on capturing the relationship between one pixel and one class, we have adapted the original LRP to capture the correlation between one patch and multiple classes.

---

> ### Author Response · Authors · 2023-11-21
> **Response to Reviewer h5Sc (part 2/2)**
>
> > Q2: As I understand, LRP in [A] requires a relevance score back-propagation for every computation unit. But, Line 9 in Algorithm 1 is not enough to explain it. I strongly recommend authors redesign Section 4 to help readers to understand LRP. Current Algorithm 1 has a lot of flaws and errors. If it is not different from [A], I recommend simplifying the explanation and focusing on explaining the inference stage.
>
> We have added Equation 7 on Appendix B, page 13 in the updated paper to explain the relevance propagation for line 9 in the Algorithm.
>
> The relevance propagation follows the Deep Taylor Decomposition [1]:
> $$
> R^j _l =\mathcal{G}\left(\mathbf{X} _l, \mathbf{W} _l, \mathbf{R} _{l+1}\right) =\sum _i x _l^j \frac{\partial L _l(\mathbf{X} _l, \mathbf{W} _l)}{\partial x_l^j} \frac{R^i _{l+1}}{L _{l}(\mathbf{X} _l, \mathbf{W} _l)}, \tag{7}\
> $$
> where $l\in$ {$1,..., L$} is the layer index in a network and $L_l(\mathbf{X}_l, \mathbf{W}_l)$ is the $l$-th layer's operation on the input feature map $\mathbf{X}_l$ with weight $\mathbf{W}_l$;  $x_l^j$ refers to the $j$-th element of the input and $R^j_l$ is the $j$-th relevance value in layer $l$.
>
> Based on this, the patch-level relevance score $R^j_{(l,c)}$ (Line 9 in the updated Algorithm 2  on page 14) of the $j$-th patch to the target class $c$ on layer $l$ (for ViT using GELU activation) can be calculated as follows [2]:
> $$
> R^j _{(l,c)} =\mathcal{G} _Q(\mathbf{X} _l, \mathbf{W} _l, \mathbf{R} _{(l+1,c)})
> =\sum _{i \mid(i, j) \in Q} \frac{x^j _l w^{j i} _l}{\sum _{ j^{\prime} \mid (i, j^{\prime} ) \in Q } x^{j^{\prime}} _l w^{j^{\prime} i} _l} R^i _{(l+1,c)},\notag
> $$
> where $l$ denotes the ViT block index, $Q:=$ {$(i, j) \mid x_l^j w_l^{j i} \geq 0$} is the subset of patch index pairs corresponding to positive relevance values and indices $i, j$ represent image patches.
>
> Thank you very much for your suggestion. We have relocated the details about using modified LRP to Appendix B on page 13.
>
> > Q3: I think LRP statistics on the training set are used for inference, like batch-norm. Am I right? It is an important point but not clearly described. Please clarify the inference process. Especially, it should be clearly stated to check whether additional computation for LRP is excluded from inference costs or not. If LRP computation is required for every inference, I will lower my rating.
>
> Thank you very much for your question. Your understanding is correct: LRP statistics are obtained **offline** (from the training dataset) and stored, thus no additional cost at inference. The only overhead lies in the fusion of intra-/inter-layer scores which is minimal (and has been factored in when measuring the throughput). We have added a paragraph at the end of Section 4 on page 4 to clarify the usage of LRP statistics during inference. Thank you once again for your question.
>
> > Q4: What is the impact of the fine-tuning process? Can STAR work without fine-tuning?
>
> We appreciate the reviewer's question. The impact of fine-tuning is to enhance a model after pruning.  We have listed the impact of fine-tuning on accuracy in the newly added Table 7, Appendix E on page 19: fine-tuning can enhance model accuracy by a minimum of 0.4% and a maximum of 4%. We have further listed how the fine-tuning is conducted in the first paragraph of Appendix E on page 16. Indeed, STAR can operate without fine-tuning. The experimental results listed in Tables 4 and 5, Appendix E on page 18 (measuring the model accuracy for variable compression ratios) are without fine-tuning.
>
> > Q5: Could you visualize saved training stats of inter-layer scores (LRP)? Is there any pattern related to CLS token?
>
> Yes, there is a relationship between the calculated patch importance and the CLS token. We have added the visualization of the saved inter-layer patch importances on Figure 1, page 4. This figure reflects the significance of each image patch to the [CLS] at the last layer and it illustrates that patch importance varies among ViT Blocks.
>
> We want to thank you for your very helpful review.
>
> [1] Explaining NonLinear Classification Decisions with Deep Taylor Decomposition, Pattern Recognition 2017
>
> [2] Transformer Interpretability Beyond Attention Visualization, CVPR 2021

---

> ### Comment · Reviewer_h5Sc · 2023-11-22
>
> Thank you for your responses.
> It cleared my questions on the paper.
>
> I will keep my initial rating.

---

### Author Response · Authors · 2023-11-21
**Common Response**

Dear Reviewers,

We would like to express our sincere gratitude for taking the time to review our paper and provide constructive suggestions. Your feedback has been invaluable in improving the content and presentation of the paper.  We have addressed all your comments in the revised paper (edits are shown in blue) and accompanying individual responses. In the following, we provide a clearer description of the main novelty points and a summary of the key changes.

**Novelty**:

- Evaluation of **inter-layer importances** of patches. This is accomplished offline using (a slight modification of) Layer-wise Relevance Propagation (LRP), a well-established tool in model interpretability. That is, the novelty is in asking the question (of inter-layer importances) rather than the tool used to solve it (LRP).
- **Fusion** mechanism for inter-layer (offline evaluation) and intra-layer (obtained at inference) importances.
- Novel **auto-tuning** mechanism for the layer-wise compression rates with no overhead of re-training: this uses offline metrics (ACS) and contains a single hyperparameter ($\rho$) to control the overall compression. Two distinctive attributes are a) establishing a *monotonicity* property and b) the use of *adaptive* (i.e., input-based) compression rate.

**Main changes:**

- **Abstract**: we have omitted some technical details regarding the "how" to improve clarity on page 1, as suggested by Reviewer kdgo.
- **Introduction**: in response to feedback from Reviewer kdgo, we have simplified the description of our contributions and summarized the novelties on page 2. Furthermore, in response to Reviewer fwdp, we have incorporated additional references on dynamic pruning methods  (Tang et al., 2021; Lin et al., 2020; Wang et al., 2020) on page 1.
- **Related work**: we clearly state the distinct focus of channel pruning and patch pruning in the second paragraph of page 3 (Reviewer fwdP). We cite and compare our method with DiffRate (as suggested by Reviewer cb4H), highlighting the differences in the third paragraph of page 3.
- **Section 4**: we have revised the content and repositioned the original Algorithm 1 to the new Appendix B on page 13 (in view of the fact that the LRP method is not one of our contributions). In the revised Section 4, we present a succinct overview of our approach and pinpoint the slight modification of the LRP adopted in our framework on page 4. Additionally, we have provided clarification on the relevance propagation step and the utilization of inter-layer importance during inference, addressing the concerns raised by Reviewer h5Sc.
- **Experiments**: we clearly state the fine-tuning and throughput calculation settings in Appendix E (page 16). Additionally, we have incorporated experiments that test the model's performance across different compression ratios and on an out-of-distribution dataset (ImageNet-R), including an analysis of the influence of data size on the pre-computed ACS, the impact of fine-tuning, and a comparative study between STAR and the new baseline DiffRate in Appendix E on pages 18 and 19 (Reviewer kdgo, cb4H, h5Sc).

Finally, we would like to thank all reviewers again for your insightful feedback.



Sincerely,

Authors of Paper 4519

---

### Meta-Review · Area_Chair_zFAH · 2023-12-05

**Metareview:**

Four experts reviewed the paper. All were supportive of the paper, but two reviewers still expressed concerns about the method's novelty and presentation clarity. The rebuttal addressed the reviewers' questions.

**Justification For Why Not Higher Score:**

Reviewers have concerns about the method novelty and presentation clarity.

**Justification For Why Not Lower Score:**

All reviewers were supportive.

---

### Decision · Program_Chairs · 2024-01-16

Accept (poster)